# Experimental investigation of wind turbine wake and load dynamics during yaw manoeuvres

Stefano Macrí[1], Sandrine Aubrun[2], Annie Leroy[1,3], and Nicolas Girard[4]

[1]Univ. Orléans, INSA-CVL, PRISME EA4229, 45072 Orléans, France
[2]Ecole Centrale de Nantes, LHEEA, 1 rue de la Noë, 44321 Nantes, France
[3]Centre de Recherche de l'École de l'Air, B.A. 701, 13661 Salon-de Provence, France
[4]Engie Digital, Le Monolithe, 59 rue Denuzière, 69285 Lyon, France

**Correspondence:** Sandrine Aubrun (sandrine.aubrun@ec-nantes.fr)

**Abstract.**

This article investigates the far wake response of a yawing upstream wind turbine and its impact on the global load variation of a downstream wind turbine. In order to represent misalignment and realignment scenarios, the upstream wind turbine was subjected to positive and negative yaw manoeuvres. Yaw manoeuvres could be used to voluntarily misalign wind turbines when
wake steering control is targeted. The aim of this wind farm control strategy is to optimize the overall production of the wind farm and possibly its lifetime, by mitigating wake interactions. While wake flow and wind turbine load modifications during yaw manoeuvres are usually described by quasi-static approaches, the present study aims at quantifying the main transient characteristics of these phenomena. Wind tunnel experiments were conducted in three different configurations, varying both scaling and flow conditions, in which the yaw manoeuvre was reproduced in a homogeneous turbulent flow at two different
scales, and in a more realistic flow such as a modelled atmospheric boundary layer. The effects of yaw control on the wake deviation were investigated by the use of stereo Particle Imaging Velocimetry while the load variation on a downstream wind turbine was measured through an unsteady aerodynamic load balance. While overall results show a non dependence of the wake and load dynamics on the flow conditions and Reynolds scales, they highlight an influence of the yaw manoeuvre direction on their temporal dynamics.

## 1 Introduction

The rising market demand for wind energy, together with the need to reduce costs and maximize power yield, have led to an increase in wind farm density (Pao and Johnson (2009)) with a concomitant increase in wake interactions. Wake interactions are the main cause of power losses and increase of fatigue loads in wind farms (Sanderse (2009)). In order to attenuate these negative effects of wind farm densification, different wake control strategies started to be envisaged. A wind farm control strategy consists in controlling each wind turbine individually in order to reduce its wake effects on the nearest downstream
turbines and maximise the total power production. The most common solutions investigated are twofold: induction control and yaw control. The former is based on a power curtailment strategy, as reducing the power extraction of an upstream wind turbine leaves more kinetic energy available for a downstream one. The latter consists in wake steering: a wind turbine (henceforth WT)

is voluntarily misaligned with respect to the wind direction in order to deviate its wake from its nominal position and hence reduce wake effects on a downstream wind turbine. If properly applied, wind farm control can improve the overall production. Its potential has already been investigated and confirmed by both simulations (e.g. Bossanyi and Jorge (2016); Gebraad et al. (2017)) and full-scale tests (e.g. Machielse et al. (2008); Wagenaar et al. (2012); Fleming et al. (2017, 2019, 2020)), although the uncertainty on the net production gain remains still significant. Some studies on the effects of yaw misalignment on wind turbine wakes, mainly based on quasi-static approaches, have already been carried out describing the effect of WT yaw on the wake position, in wind tunnel conditions (e.g. Bastankhah and Porté-Agel (2016); Grant et al. (1997); Howland et al. (2016); Schottler et al. (2018)) and at full scale (e.g Howland et al. (2020)). However, analyzing yaw manoeuvre dynamics, by studying the transient process between the non-yawed and yawed conditions, affords new insights into wake interactions. This provides the opportunity to estimate the delay between the manoeuvre on an upstream WT and its effect on a downstream WT. Analysis of the time lag between the start of yaw motion and the beginning of wake deviation effects at the downstream wind turbine position can inform us about the wake advection time. Moreover, tracking wake response during the yaw manoeuvre of a wind turbine makes it possible to compare the manoeuvre duration with the time needed for wake stabilisation in the deviated condition. The same comparisons can be made regarding the effects of the manoeuvre on the global load measured on a downstream wind turbine. Those analyses are very important to implement the proper strategy in the WT controllers. To this end, this study deals with a typical scenario of wake interaction between two wind turbines which is reproduced to investigate the effects of the yaw control strategy, paying particular attention to the WT wake and load dynamics. This work continues and completes the investigations already performed in a previous study (Macrì et al. (2018)) and aims to characterise the wake and load response to the yaw variation. It is based on an experimental approach conducted in low-speed wind tunnels. Both WTs are then modelled with porous discs. Given the low geometric scales and the focus on far-wake characteristics, it is common practice to model wind turbine rotors with porous discs (e.g. Muller et al. (2015); Yu et al. (2017); Bastankhah and Porté-Agel (2014); Van Gent et al. (2017)) to isolate the unsteady wake behavior from the wind turbine geometric and kinetic parameters. All other sources of unsteadiness (blade wakes and rotation, tip vortices, controllers) are then avoided, to focus on the wake aerodynamics only. The use of the porous disc is justified by the fact that, beside its representativeness of the wind turbine far wake (Aubrun et al. (2013); Lignarolo et al. (2016)), it is the most simplified wind turbine modelling approach and also the most used both for wake engineering models and for numerical simulations of single or multiple wind turbine wakes (see Porté-Agel et al. (2020)). Modelling a wind turbine via a porous disc implies representing a fixed operational point of a wind turbine in term of thrust coefficient and consequently velocity deficit within the wake, avoiding all the aforementioned additional sources of unsteadiness. Moreover, this approach is coherent with the general approach used for wind farm production optimisation tools (i.e. FLORIS). This kind of representation permits a good reproducibility of the results and the possibility to reproduce the far wake of a wind turbine at a low geometrical scale with a simplified model. Indeed, as experimented in Macrì et al. (2020), it is very complex to achieve satisfactory statistical reliability and reproducibility of the results obtained through the use of full scale experiments. The WT wake behaviour and the load variation on a downstream WT in interaction are investigated with particular focus on the transition between the no-yaw condition (rotor facing the wind) and the yawed condition (in this study, 30° between the wind direction and the normal to the rotor area), in both manoeuvre directions: yaw increase (0° to 30°) and

yaw decrease (30° to 0°). The same protocol is followed for each experiment, i.e. wake deviation tracking and downstream WT load variation monitoring. The wake behaviour is analyzed by the use of Particle Image Velocimetry measurements (Stereo-PIV 2D-3C) while the load variations are analyzed using a 6 Degrees of Freedom (DoF) unsteady balance. PIV fields aim at measuring the profile for the far wake velocity distribution and permit to deduce the wake deflection. The latter can be described by a wake skew angle that depends on the yaw angle and the downwind distance in particular (Bastankhah and Porté-Agel (2016)). In this study, a wake deviation angle, computed from the estimation of the wake center displacement will be considered. In addition, a wake deviation duration will be introduced to analyse the transient aspects of this wake deviation process during yaw manoeuvre. First the static wake deviation angle due to yaw misalignment and the associated global load modification on a downstream wind turbine are quantified, then the dynamical properties of the wake steering process and of the global load variation during yaw manoeuvres are studied. For this purpose, a realistic yaw manoeuvre is re-scaled in a wind tunnel to properly compare its duration with the associated wake response and load variation. In order to achieve a characterisation of this phenomenon as complete as possible, the wind tunnel experiments are conducted in three different set-ups, varying the flow conditions and the geometric scales. Two experimental campaigns are carried out in a Homogeneous and Isotropic Turbulent (HIT) incoming flow, varying the model scale and testing different flow velocities. A third experimental campaign is performed in a more realistic flow such as a modelled atmospheric boundary layer (ABL). The paper is organized as it follows. In section 2 the experimental set-ups and the data pre-processing are described. Section 3 presents the different indicators used to quantify the effects of a yawed upstream WT on the flow farther downstream, namely the upstream wake deviation, the available wind power density for a downstream WT model and the thrust applied to a downstream WT model. Static and dynamic influences of the yaw modification to these indicators are detailed in section 4 and 5, respectively. Section 6 provides a summary of the outcomes and some conclusions.

## 2 Experimental set-ups and data pre-processing

### 2.1 Wind tunnels and flow conditions

In this section the different experimental set-ups are described, then the methodology and the metrics applied to the study of the wake deviation, the available wind power variation and the disc load variation will be presented. Experiments were carried out in two wind tunnel facilities of the PRISME Laboratory at the University of Orléans. The first campaign in HIT conditions (HIT1) was performed in the Eiffel type wind tunnel, while the other two campaigns were carried out in the closed-loop wind tunnel "Lucien Malavard", respectively in the main test section for the HIT conditions (HIT2) and in the return test section for the modelled ABL conditions (ABL). The wake behaviour during yaw variation was studied in different flow conditions and scales and wind turbines were modelled with porous discs (Muller et al. (2015)) and scaled at 1:800 for the first campaign and 1:320 for the others, giving a range of Reynolds numbers based on the disc diameter of around $10^4$. Experiments were carried out with discs of two different porosity levels (henceforth P1 and P2) representing two different axial induction factors $a = \frac{1}{2}(1 - \frac{U_{wake}}{U_{ref}})$, where $U_{ref}$ is the reference wind speed at hub height and $U_{wake}$ the velocity measured within the wake. As the disc porosity is homogeneous, the velocity deficit within the near wake is characterised by a constant value that was

measured, out of the induction zone, to determine $U_{wake}$. The consequences of the yaw manoeuvre on wake steering and on the available wind power for a downstream WT were investigated at a fixed downstream distance ($\Delta x$). The fixed downstream distance was chosen according to the data set from a working wind farm studied in Garcia et al. (2019) & Macrì et al. (2020). The consequences on the aerodynamic loads applied to the downstream WT located at the same downstream distance were then performed. The wind turbine models and the spacing for the three experimental set-ups are summarized in Table 1.

**Table 1.** *Experimental set-ups.*

|       | $D$ | $\Delta x$ | $U_{ref}$ | P1 | P2 | $a_1$ | $a_2$ |
|-------|-----|-----------|-----------|----|----|-------|-------|
|       | [m] | [m] | [$ms^{-1}$] | [%] | [%] | | |
| HIT 1 | 0.1 | $3.5 \times D$ | 6 | 57 | 67 | 0.16 | 0.11 |
| HIT 2 | 0.25 | $4.2 \times D$ | 5-10-15 | 57 | 67 | 0.16 | 0.11 |
| ABL   | 0.25 | $4.2 \times D$ | 5 | 57 | 67 | 0.16 | 0.11 |

In order to ensure a relevant comparison between the results of the three campaigns, the wind turbine models and the instrumentation were installed in an equivalent configuration. The set-ups basically differ for the flow condition and/or the geometric scale, as summarized in Table.1. The upstream wind turbine model was installed in the test section at a fixed position and its yaw motion was controlled by a Kollmorgen AKM24D-ANBNC-00 rotational servomotor and measured by a Kubler type 8.5872.3832.G141 circular encoder. The disc center of this upstream wind turbine is the origin of the reference framework $x = y = z = 0$. For wake steering tracking and available wind power assessment, the velocity field was measured by stereo-PIV on a crosswise plane located at a downstream distance $\Delta x$ (Fig. 1 left). For the WT load variation measurements, a second but identical model was installed on a 6 DoF unsteady aerodynamic balance at the same downstream distance $\Delta x$ as the laser light sheet, and aligned with the upstream model (Fig. 1 right). The motion system and the acquisition board were controlled via LabView, providing a continuous measurement of the angular WT position and the load variations, when measured. More details about the measurement systems will be given in the following sections.

Regarding the flow characteristics, in HIT conditions, turbulence grids were installed at the entrance of the main test section of the two wind tunnels in order to generate the desired turbulence characteristics as in Macrì et al. (2018). The reference wind speeds $U_{ref}$, as well as the turbulence intensities in the empty field (no disc) at the disc location ($I_{Uup}$) and at the Stereo PIV measurement plane ($I_{Udown}$) are summarized in Table 2. In ABL conditions, experiments were carried out in the return test section as it was done in Muller et al. (2015) reproducing a neutrally-stratified atmospheric boundary layer (Fig. 2). The modelled ABL represents a flow above a moderately rough terrain with a roughness length of $z_0 = 5.10^{-5}$ $m$ (full scale $z_0$ = 0.02 $m$), a power law exponent of $\alpha = 0.14$ and a friction velocity of $u^* = 0.29$ $ms^{-1}$. The reference wind speed at hub height (disc center), as well as the turbulence intensities at the disc location and at the Stereo PIV measurement plane, without the presence of the disc, are also summarized in Table 2. Particular attention was paid to the yaw variation scaling in order to properly represent realistic yaw manoeuvre dynamics in a wind tunnel. Taking as full-scale references a wind speed at hub height of 12 $ms^{-1}$, a rotor diameter of $80$ $m$ and a nominal speed for yaw motion of $0.5$ $\circ.s^{-1}$, it is possible to retrieve the

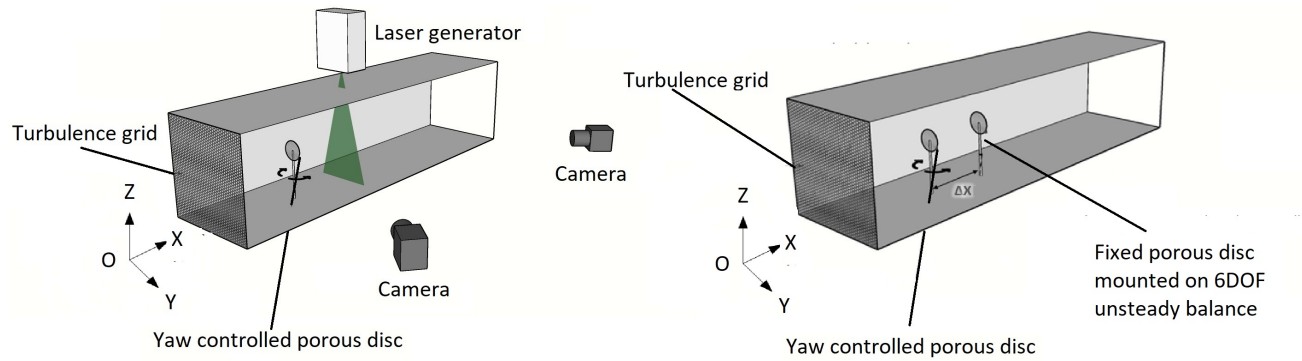

**Figure 1.** *Experimental set-ups for the 2D-3C Stereo PIV measurements (left) and for the global load variation measurements on the downstream WT (right). For both set-ups the origin O of the axis is centered on the upstream disc center.*

duration for a $0°$ - $30°$ and a $30°$ - $0°$ rotation. The yaw variation takes therefore $10\tau_0$, where $\tau_0$ is an aerodynamic time scale based on the inflow velocity at hub height $U_{ref}$ and disc diameter $D$ ($\tau_0 = \frac{D}{U_{ref}}$). For the wind tunnel condition, the yaw motion was scaled in order to have this $10\tau_0$ duration, thus respecting the Strouhal similarity based on the wind turbine rotor dimension between the reduced and full scale conditions. This similarity law is considered as the most relevant when one studies unsteady phenomena in the wake of a bluff-or porous body (Cannon et al. (1993)).

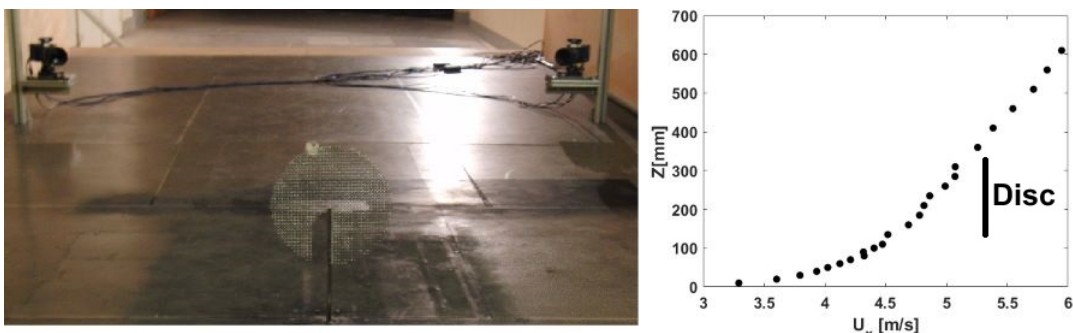

**Figure 2.** *Photo of the PIV set-up in the ABL test section of the "Lucien Malavard" wind tunnel and vertical profile of the mean streamwise velocity $U_x$.*

## 2.2   PIV measurement system

The PIV system consisted of an Nd: Yag laser Evergreen 200 ($2 \times 200 \, mJ$) emitting pulse with a wavelength $532 \, nm$, and a $2.5$ $Hz$ emission rate for non-synchronized acquisition. The light sheet was oriented in order to cross the test section transversely. Seeding particles were micro-sized olive oil droplets sprayed by a PIVTEC seeding system. Images are acquired with two LaVision Imager LX cameras ($4032 \, px \times 2688 \, px$) with a $105 \, mm$ lens equipped with a $532 \, nm$ wavelength filter. The time

delay between the images was set according to flow speeds, at 105 $\mu s$ for the HIT1 set-up, at 126, 84 and 42 $\mu s$ for the HIT2
set-up ( for $U_{ref}$ of 5, 10 and 15 $m/s$, respectively) and 126 $\mu s$ for the ABL set-up. The images were processed with a multi
pass decreasing size (64 px $\times$ 64 px, 32 px $\times$ 32 px) interrogation window with an overlap of 50 %. An ensemble-average of the
collected velocity fields is then performed. For dynamic conditions, the acquisition was triggered according to the yaw motion
progress. The collection of hundreds of image pairs was triggered at integer multiples of the time scale $\tau_0$ and a conditional
averaging approach was then applied. Indeed, due to the PIV system characteristics, during one yaw cycle from 0° to 30° and
vice versa, it was possible to acquire only one image pair at a chosen time delay. The yaw cycle was then reproduced either
300 times or 1000 times depending on the set-up (see Table 2) and a conditional averaging was applied in order to achieve the
statistical convergence of the results. By varying the time delay from 3 to $20\tau_0$, it is possible to reconstruct the phase-averaged
velocity field evolution due to the yaw motion. The maximum statistical uncertainty of the mean wind speed ($\epsilon_u = \frac{Z.I_u}{\sqrt{Nb}}$) and
its standard deviation ($\epsilon_\sigma = \frac{Z}{\sqrt{2Nb}}$ ) were defined according to Benedict and Gould (1996), with $Z = 1.96$ (confidence interval
of 95%), $Nb$ the number of independent samples, and $I_{Umax}$ the maximum turbulence intensity measured in the wake region.
These parameters, together with the model size diameter $D$, the spacing between the model and the stereo-PIV measurement
plane location $\Delta x$ and the dimensionless vector resolution $R_V$, defined as the distance between two vectors in the PIV velocity
field normalized by the disc diameters ($R_V = \Delta y/D = \Delta z/D$), are summarized for HIT and ABL conditions in Table 2. It
has to be mentioned that the present vector resolution was obtained after a bilinear interpolation on a twice finer mesh than
original PIV resolution.

**Table 2.** *Experimental configuration parameters: diameter D, spacing $\Delta x$, reference scaled wind speed $U_{ref}$, upstream turbulence intensity $I_{Uup}$, downstream turbulence intensity $I_{Udown}$, maximum turbulence intensity in the wake region $I_{Umax}$, number of samples Nb, maximal statistical uncertainty of the mean wind speed $\epsilon_u$, maximal statistical uncertainty of the wind speed standard deviation $\epsilon_\sigma$, dimensionless vector resolution $R_V$*

|  | $D$ | $\Delta x$ | $U_{ref}$ | $I_{Uup}$ | $I_{Udown}$ | $I_{Umax}$ | $Nb$ | $\epsilon_u$ | $\epsilon_\sigma$ | $R_V$ |
|---|---|---|---|---|---|---|---|---|---|---|
|  | [m] | [m] | [$ms^{-1}$] | [%] | [%] | [%] |  | [%] | [%] |  |
| HIT1 | 0.1 | $3.5 \times D$ | 6 | 4.8 | 4 | 12 | 300 | 1.6 | 9 | $8.5 \times 10^{-3}$ |
| HIT2 a | 0.25 | $4.2 \times D$ | 5 | 4.5 | 4 | 12.6 | 300 | 1.4 | 8 | $5 \times 10^{-3}$ |
| HIT2 b | 0.25 | $4.2 \times D$ | 10 | 4.5 | 4 | 10.5 | 300 | 1.2 | 8 | $5 \times 10^{-3}$ |
| HIT2 c | 0.25 | $4.2 \times D$ | 15 | 4.5 | 4 | 13.2 | 300 | 1.4 | 8 | $5 \times 10^{-3}$ |
| ABL | 0.25 | $4.2 \times D$ | 5 | 11 | 11 | 16 | 1000 | 1 | 4 | $5 \times 10^{-3}$ |

## 2.3 Load measurement system

The load measurements were performed by an unsteady 6-component aerodynamic balance $ATI^{TM} model mini 40$. The balance was mounted on a rigid structure located underneath the test section floor, and the downstream wind turbine model was installed on a specifically designed support. The balance has 6 analogic channels that, as for the encoder measuring the yaw variation, are acquired at a frequency of 2 kHz by a National Instrument card. The whole system is controlled by Labview.

It should be noted that the aerodynamic balance can detect the asymmetric loading on the downstream WT. Nevertheless the analysis will be focused on the thrust measurement because that is the only loading that can be related to a notion of WT performance. In the case of static load measurements (no yaw variation of the upstream model), the sampling time was 2 min. In the case of dynamic configurations, acquisitions were designed in order to perform cycle-averaging of load for a minimum of 500 consecutive 0°-30° backward and forward yaw displacement cycles, separated by a pause of at least the same duration as the yaw motions. A pre-processing of the acquired balance time series is needed to filter out signal fluctuations due to the natural frequency of the balance. A zero-phase digital low pass filtering strategy was applied to the balance signal in order to filter out the balance resonance without rejecting the first harmonics of the cyclic yawing frequency. The filtering tuning was driven by the duration metrics assessment (§ 5.1), in order to limit its effect on the ramp slope, despite the residual overshoot at the edges. An example of the load measurement data processing for a dynamic yaw variation is given for the HIT2 set-up in Fig.3. Figure 3a shows the power spectral density of the yaw variation and of the low-pass filtered and non-filtered model drag fluctuations. After subtracting the offset voltages measured before starting the experiment, the six low-pass filtered voltage time series were converted into the six load components via a conversion matrix provided by the balance constructor. The 6 degree-of-freedom load tensor is retrieved, but the single load of interest for the present study is the WT model drag force $F_x$. Due to the balance resolution and sensitivity, it was chosen to retrieve the drag force by the measurement of the moment along the transverse axis $T_Y$ divided by the lever arm $b$ ($F_x = T_Y/b$). By analogy with a real wind turbine, drag force will be called "Thrust" $T$ in the following.

For dynamic conditions, the last part of the data processing consists in the proper phase-averaging on the rising and descending yaw ramp. Indeed, thanks to the synchronisation of the encoder and the balance acquisitions, it was possible to isolate each ramp of yaw variation (henceforth positive yaw variation for 0° to 30°, and negative yaw variation for 30° to 0°) relying on the yaw measurement. In this way it was possible to properly and repeatably detect the start and the end of each ramp in order to make a representative phase average of all the ramps. Figure 3b shows, as example, the averaging results for a "cycle averaging" approach. The results highlight the post processing protocol efficacy of the averaging effect since the shape of the time evolution of $\frac{T}{T_{start}}$ in the absence of a filtering strategy shows that the phase-average itself is not sufficient to filter out the balance resonance. The shape of the $\left(\frac{T}{T_{start}}\right)_{filtered}$ shows the smoothing effect of the filtering strategy. Moreover, due to the general superposition of the two curves, it is assumed that the residual oscillations of the non-filtered cycle-averaged curve were related to the balance resonance and that filtering out its resonance frequency does not distort the main pattern. According to the balance specifications, the torque measurement uncertainty (95 % confidence level) is 1.75 % of the full-scale load (4 $Nm$). This is too close to the torque values measured for both porosity levels in the HIT1 configuration, and for the higher porosity level in ABL conditions. Consequently, these three configurations will not be considered in the further load analysis.

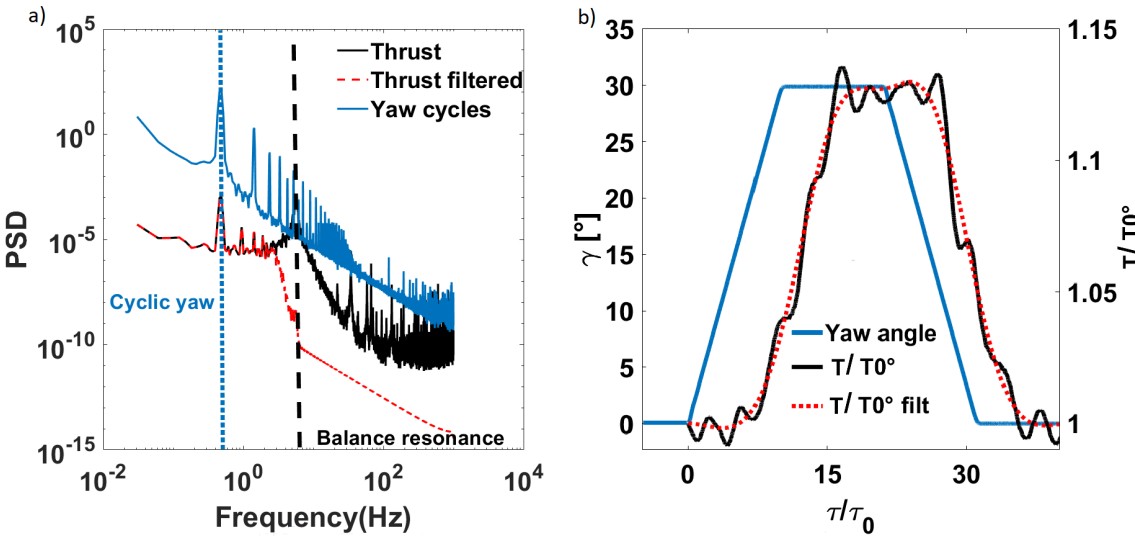

**Figure 3.** *a) Power spectral density of the yaw motion time series of 500 cycles, low-pass filtered and non-filtered drag fluctuations. b) Example of a yaw motion cycle and the associated cycle-averaged WT thrust variation*

## 3 Indicators of WT Yaw effects

### 3.1 Upstream WT Wake center position

The wake deviation was determined by estimating the displacement of the wake center position $Y_c$ from $Y_c = 0$ when the WT model is normal to the freestream flow (yaw angle equal to zero) to $Y_c \neq 0$ when the WT model is misaligned (yaw angle non null). The wake deviation angle $\theta$ can then be easily retrieved by simple trigonometric considerations (Fig. 4). These parameters are linked by the relation:

$$\chi = \gamma + \theta = \gamma + \arctan\left(\frac{\delta}{\Delta x}\right) \tag{1}$$

Where $\chi$ is the skew angle of the wake and $\gamma$ the yaw angle of the WT model. Different methods to determine the wake centre were tested ( Muller et al. (2015); Howland et al. (2016); Parkin et al. (2001); Vollmer et al. (2016); Schottler et al. (2017)), and the most robust for the present database, was the one based on the estimation of the available wind power density of a potential downstream wind turbine, as in  Vollmer et al. (2016).

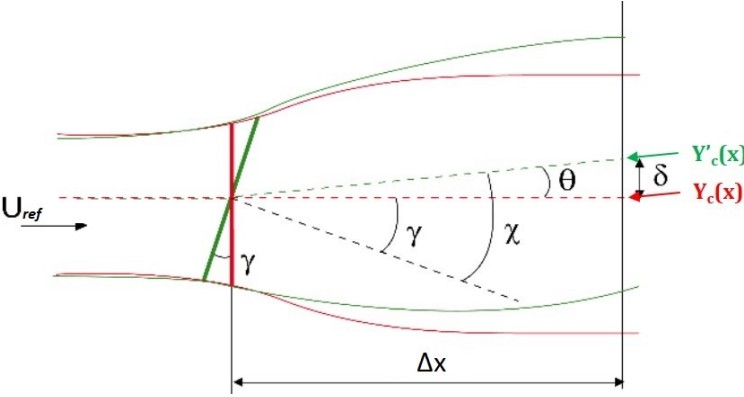

**Figure 4.** *Representation of the skew angle χ, the yaw angle γ and the deviation angle θ.*

The available wind power density ($P_{av}$ [$W.m^3.kg^{-1}$]) corresponds to the kinetic energy density crossing a virtual downstream wind turbine rotor swept area and normalized by the air density as in Eq. 2, where u is the streamwise velocity component over a plane normal to the flow direction. The available wind power density is calculated for all the possible rotor position

in the ranges of $-0.5D \leq y_0 \leq 0.5D$ and $-0.2D \leq z_0 \leq 0.2D$. The wake center position is so determined by the values of $y_0$ and $z_0$ that minimize the available power : $Y_c = y_0(minP_{av})$ and $Z_c = z_0(minP_{av})$.

$$P_{av}(\Delta x, y_0, z_0) = \frac{1}{2} \iint u^3(\Delta x, y, z) dy dz \qquad (y-y_0)^2 + (z-z_0)^2 \leq \left(\frac{D}{2}\right)^2 \qquad (2)$$

With this method, it is possible to analyze both crosswise coordinates of the wake center. Due to the negligible variation detected over the vertical coordinate $Z_c$, only the horizontal displacement of the wake will be considered. A detailed compari-

son between most of the aforementioned approaches applied to the current paper data set has been done in Macrì (2020). This comparison leaded to the choice of the method based on the available power density ( Vollmer et al. (2016)). Indeed, this, due to the integration domain definition (see above), reduces the potential sources of bias due to the PIV possible lower quality at the boarder as detailed in Macrì (2020). As concerns measurement uncertainties, taking into account the uncertainties on the wind speed, the PIV vector resolution (Table 2) and the methods used to estimate the wake center, it was possible to estimate

the maximal measurement error for both set-ups by applying usual resolution-based error estimation methods. The estimated measurement uncertainty is $\theta = \pm 0.07°$ & $Y_c = \pm 4.2 \times 10^{-4}m$ for HIT1 conditions and $\theta = \pm 0.04°$ & $Y_c = \pm 6.5 \times 10^{-4}m$ for HIT2 and ABL conditions.

### 3.2  Available wind power density at the downstream WT position

To complement the local indicator of wake deviation, which is the wake center position, the more integrated indicator provided

by the above available wind power density at the downstream WT model location $P_{avd} = P_{av}(\Delta x, y = 0, z = 0)$ was also used. This metric is interesting to analyze, because it is a good intermediate between the wake deviation of the upstream WT model

and the actual load response of the downstream WT model, when it is located at the same position as the PIV measurement plane.

### 3.3 Downstream WT thrust coefficient

The thrust coefficient $C_T$ definition is given (Eq. 3) by:

$$C_T = \frac{T}{0.5\rho U_{ref}^2 A_D} \tag{3}$$

where T is the thrust force (equivalent to the drag force in the present modelling approach) measured by the aerodynamic balance, $\rho$ the air density, $U_{ref}$ the free stream speed, $A_D$ the disc area. This parameter is a direct and reliable indicator of the dimensionless load that the model is subjected to. Moreover, since the principal aim of the study is to evaluate load variation,

it was chosen to non-dimension $C_T$ with the thrust coefficient at $\gamma = 0°$ or at the start of the yaw manoeuvre in the case of dynamic measurements.

## 4   Results for static yaw conditions

In this section the results concerning the influence of the static yaw angle applied to an upstream WT on its wake deviation, on the available wind power for a virtual downstream WT and on the actual thrust applied to a downstream WT, will be provided.

They will serve as baseline for the remainder of the study, in which dynamics will be added to the yaw motion system.

### 4.1   Wake center deviation

The measurements were performed for yaw angles from $\gamma$ between $0°$ to $30°$ by steps of $10°$. The static wake deviation was studied for all the experimental configurations listed in Table 2 and for both disc porosity levels. Figure 5a shows a summary of the wake deviation angle versus the WT yaw angle for static conditions. Several general comments can be made. Indeed,

as already shown by previous studies (Parkin et al. (2001); Howland et al. (2016); Espana (2009)), the relationship between the wake deviation angle and the yaw angle is a non-linear monotonically increasing function and the skew angle is one order of magnitude lower than the yaw angle. Theoretically, due to the absence of rotational entrainment in the wake of a porous disc and the absence of Coriolis force at such a reduced scale of observation, the absolute value of the wake deviation angle is identical for negative or positive yaw angles. A different behavior had been observed for a rotating wind turbine model, with a

light dependence of the wake deviation angle to the direction of misalignment (Bastankhah and Porté-Agel (2016); Bartl et al. (2018)). It is assumed in the present work that this asymmetry does not play a major role in the wake dynamics and therefore, is not studied. As concerns the measurement quality, the results show some scatter inherent to the propagation of cumulative errors (measurement, statistical, processing) and illustrate the difficulty of accurately determining such a small deviation angle (maximum measurement uncertainty $\theta=\pm 0.07°$). Regarding the impact of the flow conditions on the wake deviation, it is

not possible to detect any dependence of the results on flow conditions. Indeed, a scatter in values of the wake deviation

angle or the thrust is visible, but it cannot be linked to any flow conditions. It is worth to notice that the porosity level (and hence the equivalent induction factor) affects the wake deviation: cases with a lower porosity level (higher induction factor) present a higher wake deviation. In Macrì et al. (2018), present authors made some comparisons between similar experimental results and wake deviation empirical models. The trends were similar but the models systematically overestimated the wake

deviation compared to experimental values. Comparison is also very sensitive to the wake center definition used (not shown here but mentioned in Coudou et al. (2018)). Indeed, there are several wake center tracking methods and their extrapolation to skewed wakes is still under discussion. Looking at the presented results more in detail, some specific discrepancies can be stressed. Indeed, the HIT1 P2 configuration presents a higher deviation than the other ones with the same porosity level P2, without any straightforward physical ground. This can be mostly attributed to the sensitivity of the the center tracking method

as discussed in Macrì (2020). Moreover, the ABL P2 configuration presents a discrepancy between its trend (especially at $\gamma = 20°$) and the other results at the same porosity level. This is because the flow inhomogeneity together with the higher level of ambient turbulence make the velocity deficit generated by the higher porosity disc rather small and unsuitable to properly track the wake center. For these reasons, the ABL P2 configuration will not be discussed further. It is important to stress that the discrepancies that can be seen between the cases HIT2b P1 and HIT2c P1 and the rest of the cases are not unusual in

experimental measurements (Aubrun et al. (2019)) and may be due to a minor variation in the performances of one of the experimental measurement systems while changing the set-up.

## 4.2 Downstream WT thrust coefficient variation

Figure 5b shows the downstream WT thrust coefficient versus the static yaw angle applied to the upstream WT, for both porosity levels and for all the flow conditions mentioned in Table 2.

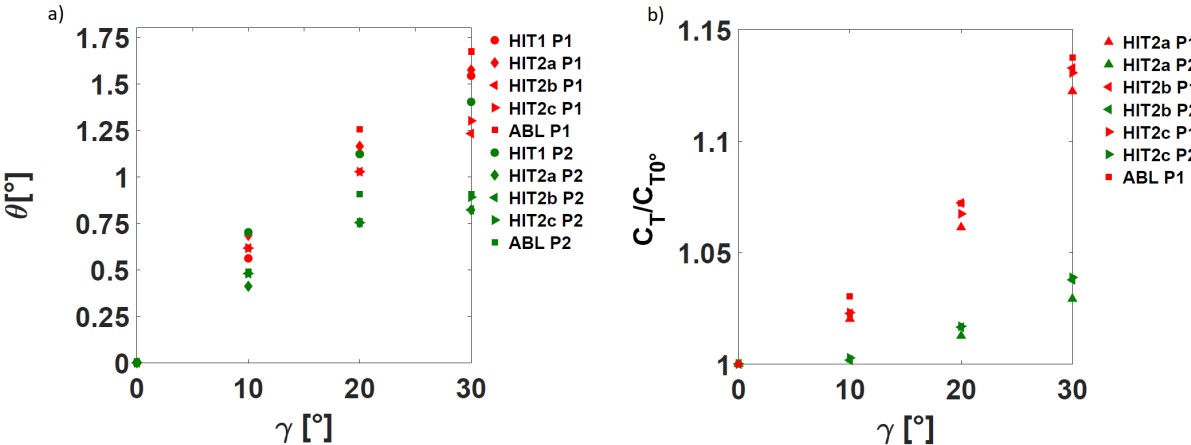

**Figure 5.** a): Wake deviation angle $\theta$ as a function of the yaw angle $\gamma$. b): Normalised thrust coefficient of the downstream wind turbine model versus the yaw angle of the upstream wind turbine. Red symbols: porosity P1, green symbols: porosity P2.

Looking at the results it is possible to state that the relationship between the downstream WT thrust coefficient and the yaw angle is a non-linear monotonically increasing function. As concerns the quality of measurements, this indicator presents less scatter than the wake deviation angle and the results do not show any unexplained outliers. Its integrative nature (global load applied to the porous disc) plays a smoothing role and the range of variations is also more significant. It is worth to notice that there is a clear thrust difference depending on the porosity level. For the P1 porosity cases, the thrust gain of the downstream WT for a 30° yaw angle variation of the upstream WT compared to the 0° case is around 13 %, irrespective of the flow conditions, while for the lower porosity level, it is about 3 %. These values suggest a remarkable influence of the porosity (and hence the axial induction factor) on the consequences of a yaw modification of an upstream WT on the load applied to a downstream one.

## 5 Results for dynamic yaw conditions

### 5.1 Metrics for dynamics

To analyse the yaw manoeuvre effects on the upstream WT wake and downstream WT thrust dynamics, some metrics to determine the properties of the transient phenomena need to be defined. It is necessary to establish a common protocol (equally reliable) to assess the transient duration, start and end for the three phenomena: yaw manoeuvre, wake deviation and thrust variation. First, the cycle-averaged yaw angle, wake deviation and thrust coefficient values were normalized in order to have transient curves between 0 and 1:

$$\Delta\gamma^* = \frac{\gamma - \gamma_{start}}{\gamma_{end} - \gamma_{start}} \tag{4}$$

with $\gamma_{start}$ and $\gamma_{end}$ the $\gamma$ values before and after the transient. It has to be noted that for positive yaw manoeuvre, $\gamma_{start} = 0$, and for negative yaw manoeuvre, $\gamma_{end} = 0$.

$$\Delta\theta^* = \frac{\theta - \theta_{start}}{\theta_{end} - \theta_{start}} \tag{5}$$

with $\theta_{start}$ and $\theta_{end}$ the $\theta$ values before and after the transient.

$$\Delta C_T^* = \frac{C_T - C_{T_{start}}}{C_{T_{end}} - C_{T_{start}}} \tag{6}$$

with $C_{T_{start}}$ and $C_{T_{end}}$ the $C_T$ values before and after the transient.

In order to facilitate the determination of the start and end of transient phenomena, fitting laws were then applied to the transient curves, depending on the yaw manoeuvre sign. For the positive yaw manoeuvre, the cycle-averaged results were fitted to Eq. 7 while for the negative yaw variation they were fitted to Eq. 8.

$$\Delta\theta^*(\tau) \text{ or } (\Delta C_T^*(\tau)) = 1 - exp\left(-\frac{\frac{\tau - \tau_{lag}}{\tau_0}}{c}\right)^3 \text{ for } \tau > \tau_{lag} \tag{7}$$

$$\Delta\theta^*(\tau) \text{ or } (\Delta C_T^*(\tau)) = exp\left(-\frac{\tau-\tau_{lag}}{\tau_0}\bigg/c\right)^3 \text{ for } \tau > \tau_{lag} \tag{8}$$

For both equations, $c$ and $\tau_{lag}$ are the fitting coefficients, $\tau$ the time triggered with the yaw manoeuvre start, and $\tau_0$ the aerodynamic time scale. $c$ represents the variation rate of the function and can be linked to the transient duration, whereas $\tau_{lag}$ represents the beginning of the function evolution and can be linked to the transient start. The fitting coefficients were determined by a classical non linear least square fitting method. Regarding the yaw angle, a fitting procedure was not necessary because of the high time resolution and precision of the acquisition. Finally, the transient start $\tau_{start}$ corresponds to the time when the curve crosses the 5% threshold of the total variation and the transient end $\tau_{end}$ to the time when the curve crosses the 95% threshold. Dimensionless values are obtained by dividing all times by the aerodynamic time scale $\tau_0$: $\tau_{start}^* = \frac{\tau_{start}}{\tau_0}$ and $\tau_{end}^* = \frac{\tau_{end}}{\tau_0}$. The transient dimensionless duration $\Delta\tau_\theta^*$ is obtained by retrieving the time difference between these two dimensionless values: $\Delta\tau_\theta^* = \tau_{end}^* - \tau_{start}^*$.

Figures 6 and 7 show examples of the transient duration determination for both wake deviation and thrust coefficient, respectively.

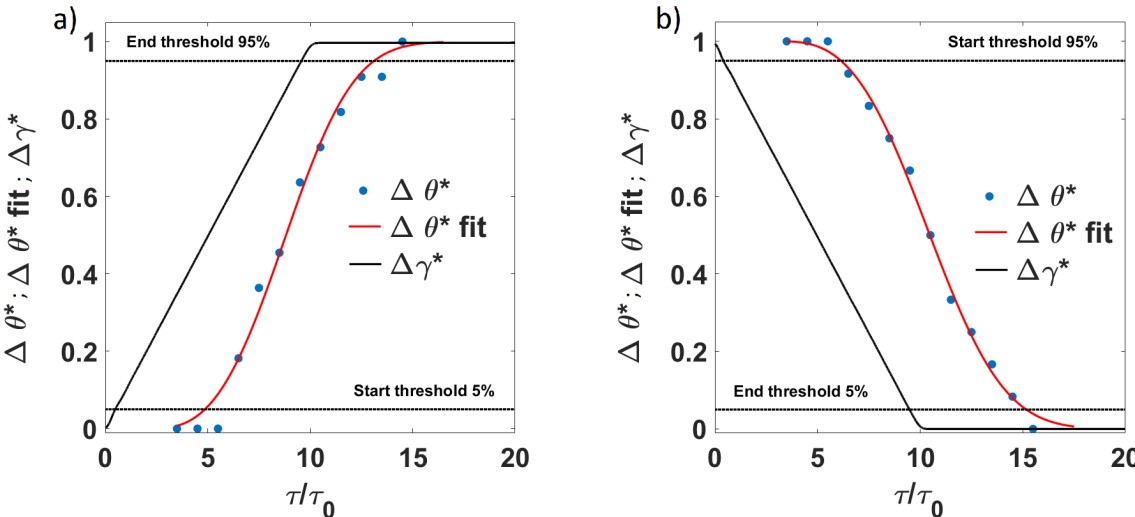

**Figure 6.** Example of cycle-averaged wake deviation history during yaw manoeuvre for configuration HIT1 P1. Positive yaw variation (a) and negative yaw variation (b). Symbols: — $\Delta\gamma^*$, ● $\Delta\theta^*$, — $\Delta\theta^*$ fitted, - - start thresholds. Data fitted for $\tau > \tau_{lag}$.

Other metrics defined for this study are $\Delta\tau_{ratio}^*$ and $\frac{U_{adv}}{U_{ref}}$. The first one represents the ratio between the wake deviation duration and the manoeuvre duration ($\Delta\tau_{ratio}^* = \frac{\Delta\tau_\theta^*}{\Delta\tau_m^*}$). The second represents the ratio between the advection velocity and the reference wind speed, where the advection velocity is defined as the ratio between the streamwise spacing $\Delta x$ between the upstream WT model and the downstream location where the wake deviation is investigated (or where the downstream wind turbine model is located) and the delay between the start of the manoeuvre and the start of the wake deviation ($U_{adv} =$

$\frac{\Delta x}{\tau_{start}}$). In order to obtain intermediate information between the upstream WT wake deviation and the response in terms of
305   thrust modification on the downstream WT, the same metrics were also calculated for the available wind power density at the
downstream WT position (see § 3.2).

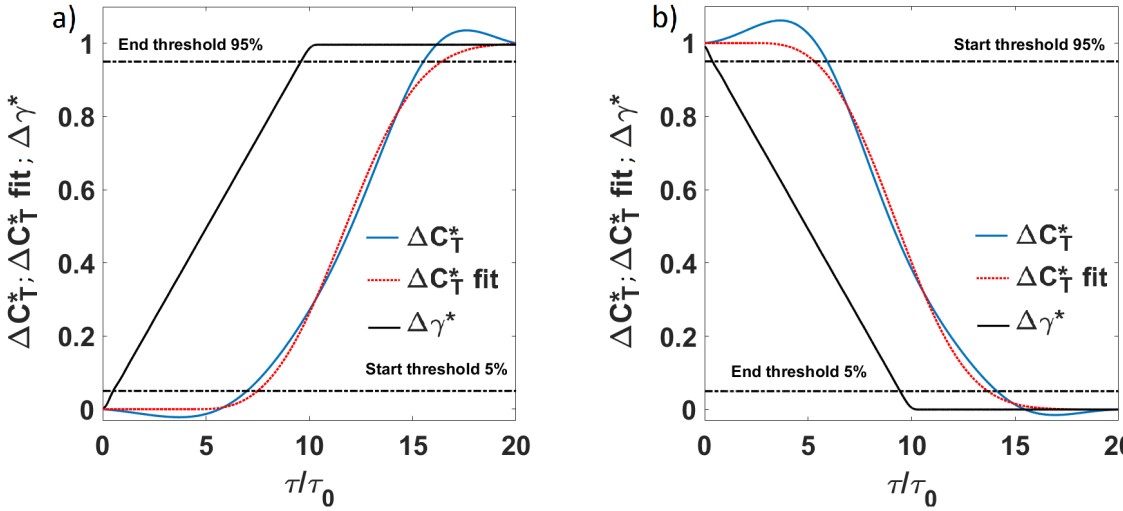

**Figure 7.** Example of cycle-averaged thrust coefficient history during yaw manoeuvre for configuration HIT2a P2. Positive yaw variation (a) and negative yaw variation (b). Symbols: — $\Delta\gamma^*$, — $\Delta C_T^*$, - - $\Delta C_T^* fitted$, - - start and end thresholds. Data fitted for $\tau > \tau_{lag}$.

All the metrics applied to the dynamic wake deviation of the upstream WT, and to the thrust variation of the downstream WT, are summarized in Tables 3 and 4, respectively. It is important to mention that for cases 10 & 11, the yaw manoeuvre speed was doubled in order to evaluate the influence of a speed-up.

**Table 3.** *Dynamic wake deviation metrics.*

| Case | set-up | | | | | | | | | | | | | |
|------|--------|---|---|---|---|---|---|---|---|---|---|---|---|---|
| | | Positive yaw manoeuvre | | | | | | | Negative yaw manoeuvre | | | | | |
| | | c | $\tau_{lag}^*$ | $\tau_{start}^*$ | $\tau_{end}^*$ | $\Delta\tau_\theta^*$ | $\Delta\tau_{ratio}^*$ | $\frac{U_{adv}}{U_{ref}}$ | c | $\tau_{lag}^*$ | $\tau_{start}^*$ | $\tau_{end}^*$ | $\Delta\tau_\theta^*$ | $\Delta\tau_{ratio}^*$ | $\frac{U_{adv}}{U_{ref}}$ |
| 1 | HIT1 P1 | 7.7 | 2 | 4 | 12.3 | 8.3 | 1 | 0.87 | 8.4 | 3 | 5.3 | 14.3 | 9 | 1.10 | 0.66 |
| 2 | HIT1 P2 | 7.9 | 0.9 | 3 | 11.5 | 8.5 | 1.03 | 1.1 | 8.5 | 2.7 | 5 | 14.1 | 9.1 | 1.11 | 0.7 |
| 3 | HIT2a P1 | 9.6 | 0.8 | 3.9 | 14.2 | 10.3 | 1.13 | 1.08 | 7.3 | 5 | 7.3 | 15.2 | 7.9 | 0.87 | 0.57 |
| 4 | HIT2a P2 | 8.5 | 1.3 | 4 | 13.1 | 9.1 | 1.01 | 1.05 | 7.2 | 4.6 | 6.9 | 14.5 | 7.6 | 0.85 | 0.6 |
| 7 | HIT2c P1 | 11.9 | 0.3 | 3.9 | 16.6 | 12.7 | 1.54 | 1.08 | 6.9 | 4.6 | 6.3 | 13.6 | 7.3 | 0.89 | 0.67 |
| 9 | ABL P1 | 8.2 | 2 | 4.6 | 13.3 | 8.7 | 0.96 | 0.91 | 7.6 | 4.4 | 6.8 | 15 | 8.2 | 0.90 | 0.61 |
| 10 | HIT2a 2S P1 | 4.8 | 3.2 | 4.6 | 9.7 | 5.1 | 1.21 | 0.91 | 3.8 | 4.7 | 5.8 | 9.9 | 4.1 | 0.97 | 0.72 |
| 11 | HIT2a 2S P2 | 4.5 | 3.4 | 4.7 | 9.4 | 4.7 | 1.13 | 0.89 | 3.9 | 4.4 | 5.5 | 9.7 | 4.2 | 0.99 | 0.76 |

**Table 4.** *Dynamic thrust metrics.*

| | | Positive yaw manoeuvre | | | | | | Negative yaw manoeuvre | | | | | |
|---|---|---|---|---|---|---|---|---|---|---|---|---|---|
| Case | set-up | c | $\tau_{lag}^*$ | $\tau_{start}^*$ | $\tau_{end}^*$ | $\Delta\tau_{C_T}^*$ | $\tau_{ratio}^*$ | c | $\tau_{lag}^*$ | $\tau_{start}^*$ | $\tau_{end}^*$ | $\Delta\tau_{C_T}^*$ | $\tau_{ratio}^*$ |
| 3 | HIT2a P1 | 8.3 | 4 | 6.9 | 15.9 | 8.9 | 0.98 | 7.8 | 2.4 | 4.8 | 13.2 | 8.4 | 0.98 |
| 4 | HIT2a P2 | 6.2 | 6.5 | 8.3 | 15 | 6.7 | 0.73 | 6.3 | 1.9 | 3.7 | 10.5 | 6.7 | 0.73 |
| 5 | HIT2b P1 | 7.7 | 5.2 | 7.2 | 15.5 | 8.3 | 0.97 | 6.9 | 3.6 | 5.3 | 12.7 | 7.4 | 0.97 |
| 6 | HIT2b P2 | 6.6 | 6.3 | 8 | 15 | 7 | 0.83 | 5.9 | 3.5 | 4.9 | 11.2 | 6.3 | 0.83 |
| 7 | HIT2c P1 | 7.4 | 5.8 | 7.6 | 15.5 | 7.9 | 0.97 | 6.7 | 3.8 | 5.4 | 12.6 | 7.2 | 0.98 |
| 8 | HIT2c P2 | 6.2 | 6.9 | 8.2 | 14.9 | 6.7 | 0.81 | 7.1 | 2.4 | 4.1 | 11.7 | 7.65 | 0.81 |
| 9 | ABL P1 | 8.7 | 4.5 | 7.2 | 16.5 | 9.3 | 1.03 | 7.9 | 2.6 | 5 | 13.4 | 8.4 | 1.03 |
| 10 | HIT2a 2S P1 | 4 | 5 | 6.1 | 10.4 | 4.3 | 1.03 | 3.1 | 4.6 | 5.4 | 8.7 | 3.3 | 1.02 |
| 11 | HIT2a 2S P2 | 4.7 | 4.3 | 5.6 | 10.7 | 5 | 1.19 | 3.8 | 2.9 | 3.9 | 7.9 | 4 | 1.18 |

## 5.2 Transient durations

The first parameter to analyze in order to characterize the dynamical consequences of the yaw manoeuvre is $\tau_{ratio}^*$, since this parameter is useful to compare the duration of the yaw manoeuvre with the duration of the induced wake flow and load modifications. Figure 8 shows an intuitive visual way of analyzing $\tau_{ratio}^*$, by plotting $\Delta\tau_\theta^*$ and $\Delta\tau_{C_T}^*$, respectively, against the manoeuvre duration $\Delta\tau_m$, values far from the diagonal line indicate a dynamic behaviour of the wake deviation or of the thrust variation different than the yaw manoeuvre. If values are below the diagonal, the response to the manoeuvre is faster than the yaw manoeuvre duration; if values are above the diagonal, the response to the manoeuvre is slower than the manoeuvre duration. The results are classified according to the disc porosity level and the yaw manoeuvre direction. In general, wake deviation durations are all around the $\tau_\theta^* = \tau_m^*$ line, except for one outlier (case 11) where the difference can be explained by a higher RMSE for the fitting procedure (due to a more scattered wake center evolution). No significant difference between the wake dynamics and the yaw manoeuvre can be noticed, whatever the disc porosity levels and the flow conditions. In contrast, thrust variation durations are in general lower than the yaw manoeuvre duration. The influence of the porosity level can be noticed, and in general, cases of negative yaw variation evolve faster than cases of positive yaw variation, especially for porosity level P1. Concerning the cases of higher yaw manoeuvre speed, no clear trend is visible on the effect of porosity and yaw manoeuvre. This may be due to the smaller number of cases analysed, but generally a higher yaw manoeuvre speed does not seem to significantly impact the behaviour of the downstream WT thrust variation.

In Tables 3 and 4, the fitting coefficient $c$ that represents the transient variation rate is given. There is a direct relationship between this parameter and the dimensionless transient durations $\Delta\theta^*(\tau)$ and $\Delta\tau_{C_T}^*$ deduced from thresholds (§ 5.1). This fitting parameter is not further exploited in the present study but its robustness suggests that it could be used to set up dynamic models of the wake deviation or of the thrust variation.

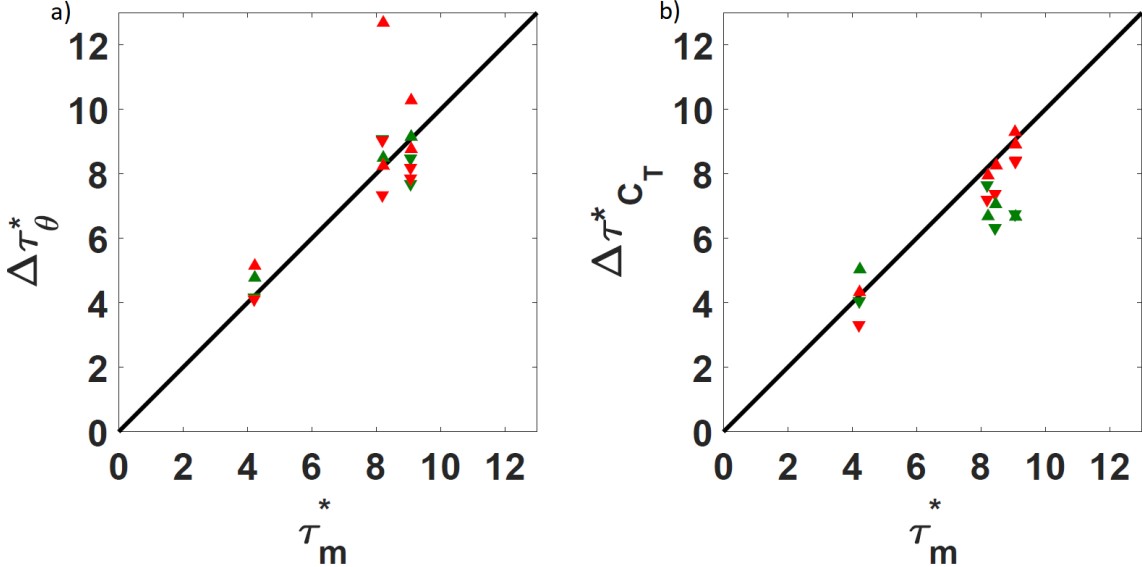

**Figure 8.** a) Wake deviation duration versus the yaw motion duration,b) Thrust variation duration versus the yaw motion duration. Summary of all the treated cases. Symbols: ▲ positive yaw manoeuvre duration, ▼ negative yaw manoeuvre duration. Colors: red for porosity P1, green for porosity P2.

### 5.3 Transient starts and ends

In order to obtain more detailed knowledge of the timing characteristics and the history of modification, it is important to check $\tau_{start}^*$ and $\tau_{end}^*$ for wake deviation, available wind power density and thrust variations, since $\tau_{start}^*$ values give information about the time delays before the upstream WT wake starts deflecting, before the available wind power density for the downstream WT is modified and then before the thrust applied to the downstream WT starts increasing. $\tau_{end}^*$ values inform about the delays necessary for the stabilisation of all the variations in the final state .

Figure 9 shows a summary of the time parameters, through a timeline representation for the wake deviation, the available wind power density and the thrust variations. The transient duration values are also shown in a parallel plot to facilitate the reading. From the analysis of the aforementioned parameters, several conclusions on the wake dynamics can be drawn. In general, the wake deviation transient (green timelines) has slightly shorter duration and starts later for the negative yaw variation than for the positive one. As explained in §5.1, a time delay between the start of the yaw manoeuvre and the wake deviation is expected since the wake deviation is observed at a certain downstream distance. Air mass needs time to travel from the upstream WT model to the downstream location of interest. This wake transport velocity, or advection velocity, is generally assessed as equal to the freestream velocity $U_{ref}$ (Trujillo et al. (2011)), but some previous studies also proposed a lower value due to the velocity deficit within the wake of $0.8U_{ref}$ (Machefaux et al. (2015)) or an average between the freestream velocity and the wake speed (Bossanyi (2018); Keck et al. (2014)). This information is checked through the ratio $\frac{U_{adv}}{U_{ref}}$ reported in Table 3. The ratio is presently close to 1 for a positive yaw manoeuvre, and around 0.8 for a negative yaw manoeuvre. A detailed

analysis of timing parameters does not provide any evidence of a major effect of the porosity on the wake deviation behaviour. Concerning the cases of higher yaw manoeuvre speed, no particular impact of the yaw manoeuvre speed on the wake response is visible. Theoretically, the fitting coefficient $\tau*_{lag}$ lag (Tables 3 and 4) can be interpreted as a time delay before the transient

350   starts. The relationship between this parameter and $\tau^*_{start}$ will be investigated in §5.4 together with the relation between $c$ and the transient duration.

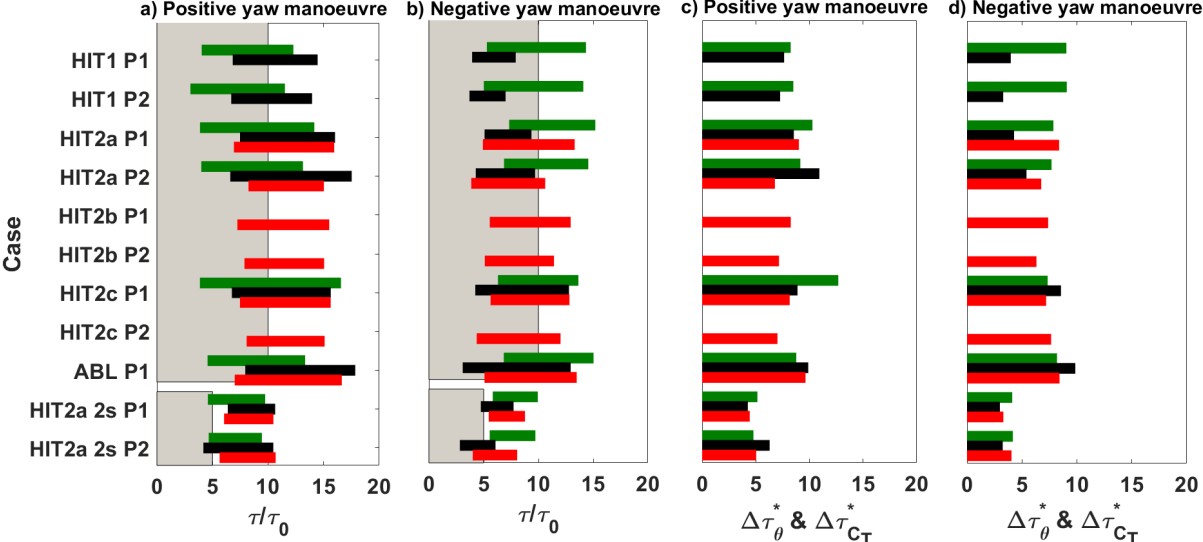

**Figure 9.** Timeline representation of the wake deviation, the available wind power density and the thrust variations for different porosity levels and flow conditions. For figures a & b the horizontal bar extremes represent the start and the end of the studied phenomenon. a) Positive yaw manoeuvre main parameters, b) Negative yaw manoeuvre main parameters, c) Positive yaw manoeuvre duration, d) Negative yaw manoeuvre duration. Colors: green = values retrieved by the center wake position, black = values retrieved by the wind power density, red = values retrieved by the thrust variations, gray = expected yaw manoeuvre duration

Figure 9 also presents the transient parameters for the available wind power density deduced from the wake measurement by PIV at the downstream location where the second WT model will be installed (black timelines). Systematic differences in the timing parameters of this available wind power density compared to the wake center deviation are visible, with shorter starting

355   time delays for the positive yaw manoeuvre and longer ones for the negative yaw manoeuvre. As this value is obtained by a space integration of the wake velocity, it illustrates that the modification of the overall wake velocity field can be different from the modification of the wake center position, which is a more local indicator and for which the range of deviation to be captured is very small. This available wind power density dynamics also illustrates how the incoming flow will dynamically impact the downstream WT model and modify its corresponding thrust. It was therefore expected that the thrust transient starts and ends

360   (red timelines) will be systematically later than the available wind power ones. The additional time delay between both would represent the WT model response to the modification of the incoming flow. This trend is not verified in the present results since

both situations can be observed. The reasoning was based on the hypothesis that the induction zone of the downstream WT model does not play a role in its dynamic process which is clearly not true.

The thrust variation starts later for a positive yaw manoeuvre than for a negative one. This trend is in opposition with the wake deviation behavior and contradicts the assumption that one can use the same time delay due to advection for the upstream WT wake deviation and for the downstream WT thrust variation.

## 5.4 Interpretation of fitting law coefficients

In this section, an evaluation of the fitting coefficients of the exponential law is done with respect to the timing parameters used for the transient analysis. In figure 10a, the wake deviation duration $\Delta\tau_\theta^*$ is plotted against the $c$ coefficient of the fitting law for both positive and negative manoeuvre and porosity levels, while in figure 10b, the same analysis is done for the thrust variation duration $\Delta\tau_{CT}^*$. Generally $c$ has a very clear correlation with the duration for both wake deviation and thrust variation cases. Indeed, it possible to retrieve by linear fitting imposing the fit law to pass through the origin, a linear relationship that links $c$ to both wake deviation and thrust variation duration. This linear fitting was done separately for the wake deviation and thrust variation but leaded to the same slope of $0.93$:

$$c = 0.93\Delta\tau^* \tag{9}$$

where $\Delta\tau^*$ is either the wake deviation duration $\Delta\tau_\theta^*$ either the thrust variation duration $\Delta\tau_{CT}^*$. These results confirm the robustness of the fit coefficient $c$ although the reason that keep $c$ constantly lower than the phenomenon duration has still to be investigated. In figure 11a, the start of the wake deviation $\tau_{start}^*$ is plotted against the $\tau_{lag}^*$ coefficient of the exponential fitting laws for both positive and negative manoeuvre and porosity levels. In figure 11b, the same analysis is done for the start of the thrust variation. Several considerations on the $\tau_{lag}^*$ coefficient can be done. Indeed, although it can be observed an higher scatter than for the $c$ coefficient, $\tau_{lag}^*$ also shows a detectable trend. The thrust data seems to have lower scatter than the wake deviation data and this could be related to the higher time resolution of the load measurements. There is generally a bias between the values that oscillate between 1 and 2 $\tau_0$. This could be partially attributed to the threshold value used for the $\tau_{start}^*$ determination.

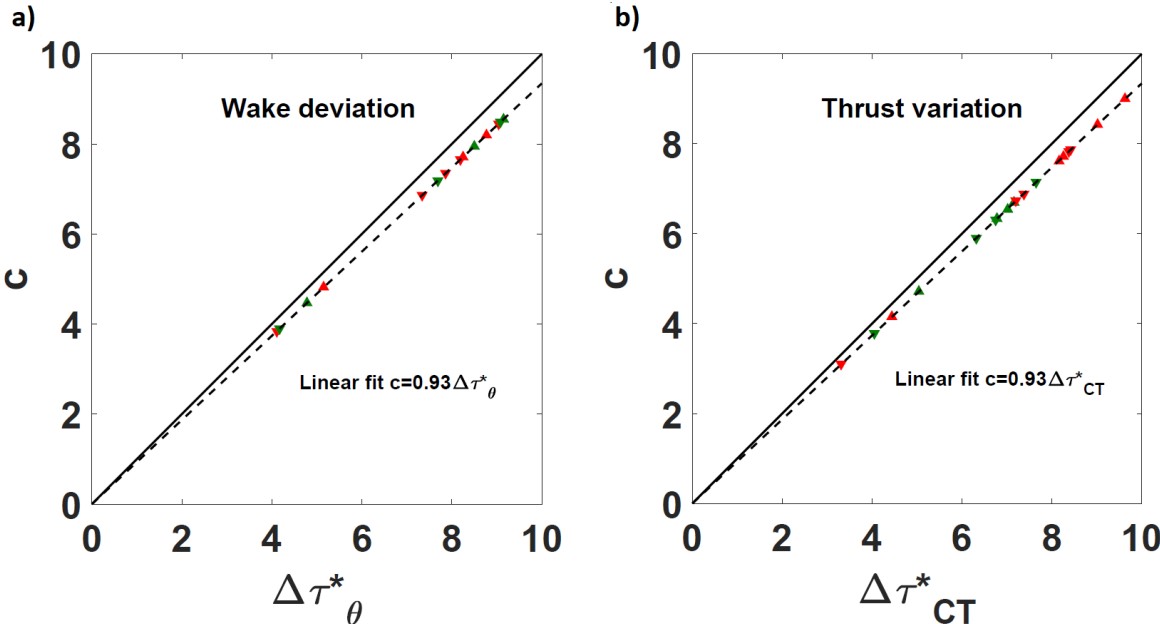

**Figure 10.** a) Wake deviation duration $\Delta\tau_\theta^*$ versus the $c$ fitting coefficient of the exponential laws. b) Thrust variation duration $\Delta\tau_{CT}^*$ versus the $c$ fitting coefficient of the exponential fitting laws. Summary of all the treated cases. Symbols: ▲ positive yaw manoeuvre duration, ▼ negative yaw manoeuvre duration. Colors: red for porosity P1, green for porosity P2.

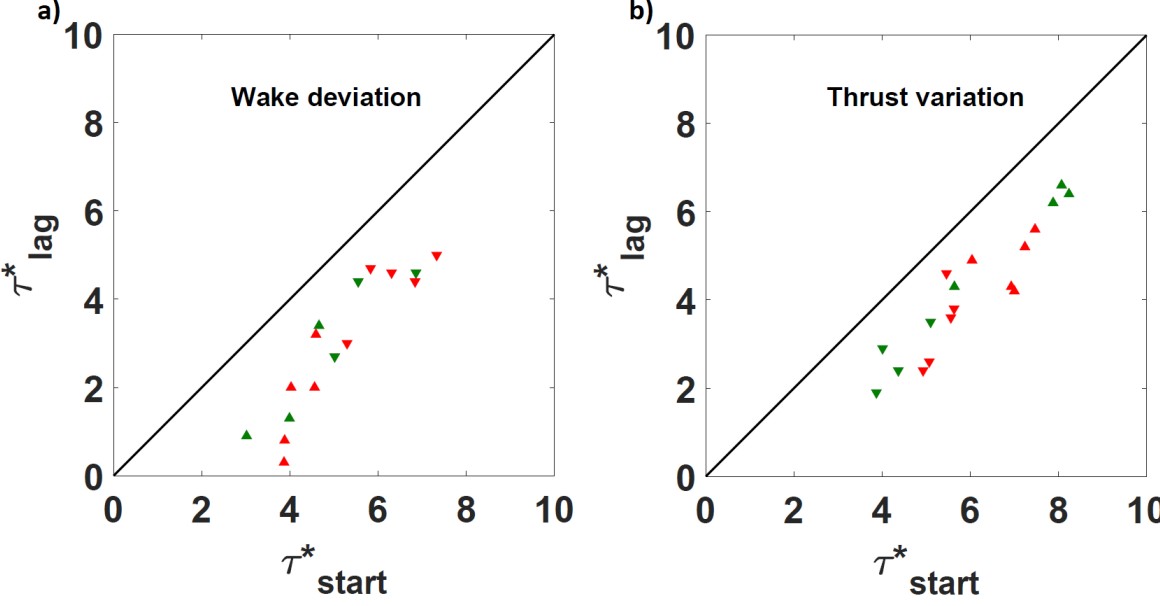

**Figure 11.** a)Wake deviation start $\tau_{start}^*$ versus the $\tau_{lag}^*$ fitting coefficient of the exponential laws. b) Thrust variation start $\tau_{start}^*$ versus the $\tau_{lag}^*$ fitting coefficient of the exponential laws. Summary of all the treated cases. Symbols: ▲ positive yaw manoeuvre duration, ▼ negative yaw manoeuvre duration. Colors: red for porosity P1, green for porosity P2.

In conclusion, the fitting laws implemented for the dynamic yaw conditions seem to be exploitable for a generalisation of a law describing the transition phenomenon. Especially, the $c$ coefficient has proven to be quite robust to be used in an empirical law to model the transient. Regarding the $\tau_{lag}^*$ coefficient, although its physical signification is clear, its implementation in an empirical law demands more caution. Indeed, the measured values do not follow a trend as clear as the one observed for the duration coefficient. This is probably due to the specificity of this parameter which is directly linked to the start of the

phenomenon, and consequently it is more sensible to dispersion. Nevertheless, $\tau_{lag}^*$, being representative of advection, could be adjusted by making some simple advection hypothesis. The higher robustness of the $c$ parameter, being representative of the transient duration, can probably due to the fact that it damps better the scatter of $\tau_{start}^*$ and $\tau_{send}^*$ which are generally concordant (both either overestimated or underestimated). At this stage, further exploitation of these data and studies have to be done in view of the implementation of an empirical model.

## 6  Conclusions

The aerodynamic characterisation of the wake of a wind turbine model and its effect on the load of a downstream similar wind turbine model consequent to a dynamic positive (misalignment scenario) or negative (realignment scenario) yaw variation were experimentally studied for different incoming flow conditions, Reynolds scales and induction factors by using porous discs. Wake deviation, available wind power or thrust variations were the main metrics estimated from PIV and aerodynamic load

measurements. First, the wake deviation angle and the thrust coefficient of the downstream wind turbine model were analysed as a function of different yaw angles in static conditions. Then, the duration, start and end of the temporal response of the metrics to a dynamic yaw variation were estimated and compared. The main results are summarized below. Overall results do not show any noticeable influence of the flow conditions (Homogeneous Isotropic flows or Atmospheric boundary layer flow), or of the Reynolds scales on the static and dynamical properties of the different metrics. The influence of the degree of

physical modelling of the wind turbine (porous disc versus rotating wind turbine model) on the results had not been studied since it was assumed that this feature does not play a major role in the yawed far-wake dynamics, but this question needs to be further investigated. Concerning the characterisation of the magnitude of the wake deviation in static yaw conditions, results show that the wake deviation is one order of magnitude lower than the yaw increment and that the relationship between the wake deviation and the yaw angles is non linear as previously found in the literature. In the case of a higher porosity level the

thrust increment is much lower than in the case of lower porosity. However, while a relevant influence of the porosity on the wake deviation angle and thrust magnitude of the downstream wind turbine model is found in static conditions, no significant influence of porosity is observed in dynamic yaw variation conditions. On the other hand, in these conditions, the analysis of the three metrics reveals different temporal characteristics depending on whether yaw variation is positive or negative. In general, the wake deviation transient has slightly shorter duration and starts later for the negative yaw variation than for the positive one.

A proper knowledge of such effects could help to determine the time necessary for the wake steering to totally develop. On the contrary, the thrust variation starts later for the positive yaw manoeuvre than for the negative one. The influence of the yaw manoeuvre speed was tested and doubling the yaw manoeuvre speed does not seem to influence the wake or load dynamics.

Finally, the study shows that the same dynamical properties of the wake of the upstream wind turbine and load variation of the downstream wind turbine cannot be generalised to any yaw variation configurations, and that the advection velocity should be

assumed to be different according to the yaw manoeuvre direction. A first step in proposing fitting coefficients was made in order to support a dynamic model of the wake deviation or the thrust for positive or negative yaw variations. However, further development will be performed to confirm the reliability of the proposed models.

*Author contributions.* SA, AL and NG co-supervised the PhD work of SM. SM contributed to the design of the experimental set-ups, conducted all experiments, processed and analysed all data. SM, SA, AL and NG contributed to the interpretation of the results. SM, SA and

AL contributed to the manuscript writing.

*Competing interests.* The authors declare that they have no conflict of interest

*Acknowledgements.* The present work is part of a PhD co-funded by Labex Caprysses and Engie.

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
