# Peer review of "Experimental investigation of wind turbine wake and load dynamics during yaw manoeuvres"

_Wind Energy Science, 2020_

## Referee Comment (RC1) · Anonymous Referee #1 · 26 Oct 2020

**REVIEW OF WES-2020-105**

*authors:*
Stefano Macrí
Sandrine Aubrun
Annie Leroy
Nicolas Girard

*Experimental investigation of wind turbine wake and load dynamics during yaw manoeuvres*

**Summary:**

The work presented in the manuscript describes metrics for a few key phenomena of interest under the dynamic variation of a wind turbine's yaw position. The subject is an important one that is likely to receive more attention in the coming years, as wake steering and active wind plant controls become more common. The main takeaways from the work seem to be that wake center and wind turbine thrust values are invariant to operating conditions, but that some of the characteristic time scales exhibit hysteresis with positive and negative yaw dynamics. The work is interesting, but requires some additional clarification before being appropriate to publish. See comments below.

**Major points:**

- page 2 – The authors justify the use of porous discs in order to eliminate many of the aerodynamic phenomena in the wakes related to the rotation and geometry of the rotor blades. Additional explanation is needed to understand how well the results presented in the paper reflect real wind turbine wakes. At the bottom of page 4, "the similarity law with the full scale condition" does not discuss the customary dimensionless parameters for dynamic scaling. Please explain how results from this work apply to real wind turbines.
- – "Time delays, multiples of the aerodynamical time scale $\tau_0$ were applied and a conditional averaging of the collected velocity fields was then performed" This requires further explanation, does this mean only a single PIV image pair was collected for each dynamic yaw maneuver or that images were taken at integer multiples of the time scale?
- Not clear how HIT2 cases are different except for max measured TI in the wake. Are these just different ensembles of observations from the same case?
- page 9 – Does the formula in Eq. (2) take into account the fact that the projected area of the rotor disc perpendicular to the flow changes with yaw angle?
- On page 9, the authors state that "Theoretically, due to the absence of rotational entrainment in the wake of a porous disc, the absolute value of the wake deviation angle is identical for negative or positive yaw angles." This argument does not take into account the coriolis force for wind turbines operating in the atmospheric boundary layer.

- In Section 4, results are presented as a list, rather than a single continuous and coherent narrative. This should be changed.
- Authors state that "No dependence on flow conditions can be detected." in Section 4 when there are obvious differences as shown in Figure 5. This statement requires additional clarification.
- Sensitivity of estimated wake center to identification methodology is a well known issue in the wind energy research community. Given this, presenting the results of only one method are questionable, especially in the case of outliers. A figure showing the estimated wake center locations with different methods would be helpful. Also, the vertical component of the wake center should be included for completeness. See Eliot Quon. SAMWICH Box: A Python-Based Toolbox for Simulated And Measured Wake Identification and CHaracterization. https://github.com/ewquon/waketracking
- "The HIT1 P2 configuration presents a higher deviation than the other ones with the same porosity level, without any straightforward physical grounds." This point is unclear. Do the authors mean that greater than other tests with the P2 mesh or than all tests with the P1 mesh? Is this evidence of the sensitivity of the center tracking method?
- The authors make a point, "The ABL P2 configuration presents a discrepancy between its trend (especially at $= 20°$) and the other results at the same porosity level. This is because the flow inhomogeneity together with the higher level of ambient turbulence make the velocity deficit generated by the higher porosity disc rather small and unsuitable to properly track the wake center. For these reasons, the ABL P2 configuration will not be discussed further." Is this discrepancy within the uncertainty bands of the measurements? The modeled ABL seems like the most meaningful and representative case tested in this work to the wakes of real wind turbines. These data should be kept and discussed in the context of the rest of the analysis.
- Last point in Section 4.1 – it is not clear to which discrepancy the authors refer. Data in figure 5a for HIT2bP1 and HIT2cP1 appear very similar.
- Last point in Section 4.2 – Authors state that "the thrust gain for a 30ĉirc yaw angle compared to the 0ĉirc case is around 13%, irrespective of the flow conditions" is a bit misleading. It is not immediately clear that the thrust gain and the yaw angle are for different turbines. This is also true for the right subfigure of Figure 5. The axis labels are a bit confusing. at first glance, it looks like a ratio of the thrust coefficient of a wind turbine under yaw to that of the same turbine without yaw. Update labels to make more clear.
- Figure 5 – These figures would benefit from error bars. Maybe place cases side-by-side in groups for each value of $\gamma$?
- Equation 4 – For clarity, authors should specify that for positive yaw maneuvers, $\gamma_{start} = 0$, and for negative yaw maneuvers, $\gamma_{end} = 0$
- It would be informative to see higher order statistics of the key phenomena of interest. PDFs of wake deviation angle and thrust coefficient would help readers and researchers understand that these are stochastic quantities.
- If the fit functions in Eqs. (7) and (8) are applied to the estimated yaw center and change in thrust of the downstream turbine, shouldn't $\tau_{lag}$ represent the convective time between the upstream turbine and either the measurement location or the downstream turbine? Also, the transient dimensionless duration is missing 10% of the fit, and thus will systematically underestimate the actual duration. This will lead to misleading values of $\Delta\tau^{\star}_{ratio}$.
- The measured overshoot of thrust coefficient is not modeled or discussed at all (see Figure 7). This seems like an important physical element of this phenomenon. Consider modifying the models in Eqs. (7) and (8) if the overshoot appears consistently. Are the curves of $C_T$ in Figure 7 averages over many maneuvers?
- Tables 3 and 4 – Do these values represent least-squares fits to average values, or are the instantaneous data fit and fit values averaged afterward? The tables would benefit from higher-order statistics or measurements of uncertainty.

- page 14, line 288 – I think $\tau_m$ should be $\Delta\tau_m$
- Figure 8 needs legends and should use different symbols as figures above, as the points represent different groupings of the test cases.
- Authors state that "Theoretically, the fitting coefficient $\tau^*_{lag}$ lag (Tables 3 and 4) can be interpreted as a time delay before the transient starts. Unfortunately, there is no clear relationship between this parameter and the $\tau^*_{start}$ start. This illustrates the difficulty of capturing the actual transient start for the present study." It seems like $\tau_{start}$ should be $\tau_{lag}$ plus the time required for the function to reach 0.5 based on the parameter $c$.

**Minor points:**

- page 1 – paper
- page 1 – which is increasingly studied
- page 1 – started to be envisaged.
- page 1 – Remove "ity"
- page 1 – Remove the work "in" from XXXX
- page 6 – Table 2 – units should be written in normal text rather than math font.
- page 6 – Remove ical
- page 6 – kHz
- page 6 – "performed in order to perform" is redundant. Please rephrase.
- page 6 – Text subscripts should be changed (e.g. $T_{\mathrm{filtered}}$ instead of $T_{filtered}$)
- page 7 – Assumptions are made by the authors in fact. Replace "can be" with "is".
- page 8 – treated – considered?
- page 8 – it is not clear what 'usual error propagation methods' are. Please be specific.

---

## Referee Comment (RC2) · Anonymous Referee #2 · 23 Nov 2020

The paper provides an experimental study of certain aspects of yaw dynamics. The experimental approach seems fine and the writing is generally clear. My main concerns lie in the contribution of the work in terms of knowledge about what the authors call wake deviation dynamics and the loading on downstream turbines, which does not seem to be reported (discussed). The conclusions and discussion of the results lack a connection to the physics, which would potentially yield the insights that the paper aims to provide.

I was confused by the term wake deviation (dynamics) as it is not precise and was not really clarified in the paper.

[Figure]

It was not clear that the balance would detect asymmetric loading on the disk, which might be important in this context. In general little discussion of the downstream turbine was affected is provided.

There are some additional yaw models not discussed in the literature review that should be included, particularly in terms of computing the behavior of the wake centerline and in capturing the yaw dynamics (which the authors claim has never been done). Given those works it is unclear that the wake deviation angle is a meaningful metric since the centerline is not a straight line due to the curled wake etc., discussed even in some of the papers cited.

I was somewhat confused by the choice of metrics in general, particularly in section 5.1. There was little discussion given as to why these are the correct metrics to consider in practice. The chosen ones for example do not provide information about what the downstream turbine might see in terms of velocity, wake or loading (although these things were alluded to in the abstract). These issues are of great importance and of far greater interest than the effect of cyclic yawing behavior, which is not clear is common in practice.

The choice of configuration was also strange since the streamwise spacing was short for all cases (in practical wind farms spacings are often at least 5-7D) and there was little justification for these choices.

The list of results in bullet form was rather superficial and were merely observations of the data rather than an analysis.

Why are equations (7) and (8) the appropriate choice; they just empirical fits to a chosen shape but the significance of this shape or choice is not mentioned.

A few minor comments

Using S for area is unnecessarily confusing

Figure 1 should be larger (there is a lot of white space so it should be easy to make it

easier to read and see all of the detail within the sketch).

---

## Author Comment (AC1) · 19 Jan 2021

Dear referee 2, We, co-authors of the present manuscript, are currently preparing their final response to your comments and we would like to ask you for additional information regarding your following comment : "There are some additional yaw models not discussed in the literature review that should be included, particularly in terms of computing the behavior of the wake centerline and in capturing the yaw dynamics (which the authors claim has never been done)." Indeed, we might not have been fully exhaustive in our literature review but it is not clear to us which missing papers that you would like to see added. Would it be possible to indicate us explicitly the references that

you had in mind? Thank you very much in advance Best regards Sandrine Aubrun, on behalf of the co-authors

---

## Author Comment (AC2) · 4 Feb 2021

**Reviewer comments**

Stefano Macrì, Sandrine Aubrun, Annie Leroy, Nicolas Girard

Co-authors thank the reviewers for their fruitful comments and will give point by point answers below. In this response to the reviewers the response of the authors not included in the revised manuscript will be written in blue while the part included in the revised manuscript will be written in red.

**1 Response to reviewer 1**

Summary:

The work presented in the manuscript describes metrics for a few key phenomena of interest under the dynamic variation of a wind turbine's yaw position. The subject is an important one that is likely to receive more attention in the coming years, as wake steering and active wind plant controls become more common. The main takeaways from the work seem to be that wake center and wind turbine thrust values are invariant to operating conditions, but that some of the characteristic time scales exhibit hysteresis with positive and negative yaw dynamics. The work is interesting, but requires some additional clarification before being appropriate to publish. See comments below.

Major points:

– page 2 - The authors justify the use of porous discs in order to eliminate many of the aerodynamic phenomena in the wakes related to the rotation and geometry of the rotor blades. Additional explanation is needed to understand how well the results presented in the paper reflect real wind turbine wakes. At the bottom of page 4, "the similarity law with the full scale condition" does not discuss the customary dimensionless parameters for dynamic scaling. Please explain how results from this work apply to real wind turbines.

The porous disc is commonly accepted in literature as a gold standard to model wind turbines at small scale in wind tunnel as well as for wake engineering models and numerical simulations of single or multiple wind turbine wakes. The far-wake of a porous disc has been proven to be similar to the one of a rotating wind turbine (Aubrun et al. (2013); Lignarolo et al. (2016)). Nevertheless it is true that these assumptions might benefit of a more complete explanation in the paper. For these reasons the following explanation regarding the statement in page 2, has been added to the revised version of the paper:

The use of the porous disc is justified by the fact that, beside its representativeness of the wind turbine far wake (Aubrun et al. (2013); Lignarolo et al. (2016)), it is the most simplified wind turbine modelling approach and also the most used both for wake engineering models and for numerical simulations of single or multiple wind turbine wakes (see Porté-Agel et al. (2020)). Modelling a wind turbine via a porous disc implies representing a fixed operational point of a wind turbine in term of thrust coefficient and consequently velocity deficit within the wake, avoiding all the aforementioned additional sources of unsteadiness. Moreover, this approach is coherent with the general approach used for wind farm production optimisation tools (i.e. FLORIS). This kind of representation permits a good reproducibility of the results and the possibility to reproduce the far wake of a wind turbine at a low geometrical scale with a simplified model. Indeed, as experimented in Macrì et al. (2020), it is very complex to achieve satisfactory statistical reliability and reproducibility of the results obtained through the use of full scale experiments.

Regarding the statement in page 4, a sentence has been added to justify our choice:

For the wind tunnel condition, the yaw motion was scaled in order to have this $10\tau_0$ duration, thus respecting the Strouhal similarity based on the wind turbine rotor dimension between the reduced and full scale conditions. This similarity law is considered as the most relevant when one studies unsteady phenomena in the wake of a bluff-or porous body (Cannon et al. (1993))

– -"Time delays, multiples of the aerodynamic time scale $\tau_0$ were applied and a conditional averaging of the collected velocity fields was then performed" This requires further explanation, does this mean only a single PIV image pair was collected for each dynamic yaw maneuver or that images were taken at integer multiples of the time scale?

The statement means that the collection of hundreds of image pairs was triggered at integer multiples of the time scale $\tau_0$ and a conditional averaging approach was then applied. Indeed, due to the PIV system characteristics, during one yaw cycle from 0° to 30° and vice versa, it was possible to acquire only an image pair at a chosen time delay. The yaw cycle was then reproduced hundred of times and a conditional averaging was applied in order to achieve the statistical convergence of the results. The statement has been modified as it follows.

The collection of hundreds of image pairs was triggered at integer multiples of the time scale $\tau_0$ and a conditional averaging approach was then applied. Indeed, due to the PIV system characteristics, during one yaw cycle from 0° to 30° and vice versa, it was possible to acquire only one image pair at a chosen time delay. The yaw cycle was then reproduced either 300 times or 1000 times depending on the set-up (see table 2) and a conditional averaging was applied in order to achieve the statistical convergence of the results.

- Not clear how HIT2 cases are different except for max measured TI in the wake. Are these just different ensembles of observations from the same case?

  The HIT2 cases differ with regard to the reference wind speed $U_{ref}$. Indeed, the aim was to check whether the absence of Reynolds effect observed for static yaw conditions is also observed in the case of dynamic yaw manoeuvre. For this reason the same experiments, with the same discs and set-up, were reproduced by only varying the reference wind speed. The observed variation of the maximal measured TI is not considered as significant, it can indeed be due to the statistical uncertainty of the wind speed standard deviation.

- page 9 - Does the formula in Eq. (2) take into account the fact that the projected area of the rotor disc perpendicular to the flow changes with yaw angle?

  Probably the reviewer refers to Eq. (3) that describes the thrust coefficient determination for the downstream porous disc. As this disc is not subject to yaw manoeuvre, its surface is always considered as normal to the main freestream flow.

- On page 9, the authors state that "Theoretically, due to the absence of rotational entrainment in the wake of a porous disc, the absolute value of the wake deviation angle is identical for negative or positive yaw angles." This argument does not take into account the coriolis force for wind turbines operating in the atmospheric boundary layer.

  Due to the low scale of the experiments, the Coriolis force is negligible/absent. The statement has been modified as it follows:

  Theoretically, due to the absence of rotational entrainment in the wake of a porous disc and the absence of Coriolis force at such a reduced scale of observation, the absolute value of the wake deviation angle is identical for negative or positive yaw angles.

- In Section 4, results are presented as a list, rather than a single continuous and coherent narrative. This should be changed.

  Results in section 4 have been modified as demanded by the reviewer. The part rearranged from the original list format will be written in green.

- Authors state that "No dependence on flow conditions can be detected." in Section 4 when there are obvious differences as shown in Figure 5. This statement requires additional clarification.

The statement referred to the absence of a clear correlation between the measurements and the flow conditions. There is a clear scatter in values but it cannot be linked to any flow conditions. The statement has been modified as it follows:

Regarding the impact of the flow conditions on the wake deviation, it is not possible to detect any dependence of the results on flow conditions. Indeed, a scatter in values of the wake deviation angle or the thrust is visible, but it cannot be linked to any flow conditions.

– Sensitivity of estimated wake center to identification methodology is a well known issue in the wind energy research community. Given this, presenting the results of only one method are questionable, especially in the case of outliers. A figure showing the estimated wake center locations with different methods would be helpful. Also, the vertical component of the wake center should be included for completeness. See Eliot Quon. SAMWICH Box: A Python-Based Toolbox for Simulated And Measured Wake Identification and CHaracterization. https://github.com/ewquon/waketracking

It is true that the reader could benefit of a comparison between different methods. Given the necessity to detail the approach of each compared method, it was chosen to refer to other published works. Anyway, these wake center tracking method comparisons, as well as the evaluation of the wake displacement in the vertical direction, have been performed on the same data set in Macrì (2020). These information have been added to section 3.1 after the statement "Due to the negligible variation detected over the vertical coordinate $Z_c$, only the horizontal displacement of the wake will be treated" as it follows:

A detailed comparison between most of the aforementioned approaches applied to the current paper data set has been done in Macrì (2020). This comparison leaded to the choice of the method based on the available power density ( Vollmer et al. (2016)). Indeed, this, due to the integration domain definition (see above), reduces the potential sources of bias due to the PIV possible lower quality at the boarder as detailed in Macrì (2020)) .

Concerning the suggested Toolbox, most of the approaches are based on Gaussian wake shapes and so not applicable to the shape of the velocity profiles of the present study, or they use an equivalent approach to the one already tested and described in Macrì (2020).

– "The HIT1 P2 configuration presents a higher deviation than the other ones with the same porosity level, without any straightforward physical grounds." This point is unclear. Do the authors mean that greater than other tests with the P2 mesh or than all tests with the P1 mesh? Is this evidence of the sensitivity of the center tracking method?

The authors mean greater than other tests with the P2 mesh, and yes it is evidence of the sensitivity of the center tracking method. The sentence has been modified as it follows:

The HIT1 P2 configuration presents a higher deviation than the other ones with the same porosity level P2, without any straightforward physical ground. This can be mostly attributed to the sensitivity of the the center tracking method as discussed in Macrì (2020).

– The authors make a point, "The ABL P2 configuration presents a discrepancy between its trend (especially at = 20°) and the other results at the same porosity level. This is because the flow inhomogeneity together with the higher level of ambient turbulence make the velocity deficit generated by the higher porosity disc rather small and unsuitable to properly track the wake center. For these reasons, the ABL P2 configuration will not be discussed further." Is this discrepancy within the uncertainty bands of the measurements? The modeled ABL seems like the most meaningful and representative case tested in this work to the wakes of real wind turbines. These data should be kept and discussed in the context of the rest of the analysis.

It is true that the modeled ABL is the most representative of real wind turbine wakes. Unfortunately, for the higher porosity P2, the data set has not a satisfactory quality to be considered as exploitable. Indeed, the discrepancy is higher than the uncertainty bands of the measurements. It was chosen to mention this in the paper to give an indication about the possibility (or not) to properly track the wake of a porous disc in similar conditions with a similar set-up (porosity level, PIV system and flow conditions). Nevertheless, the ABL case is fully discussed for the porosity P1, and from this analysis, it is possible to conclude that the presence of the ABL does not impact significantly the wake behaviour as concerns static and dynamic yaw manoeuvre.

– Last point in Section 4.1 - it is not clear to which discrepancy the authors refer. Data in figure 5a for HIT2bP1 and HIT2cP1 appear very similar.

The authors refer to the discrepancies between HIT2bP1 and HIT2cP1 and the rest of the cases. It is true that the statement is a bit misleading. The statement has been modified as it follows:

The discrepancies that can be seen between the cases HIT2b P1 and HIT2c P1 and the rest of the cases are not unusual in experimental measurements (Aubrun et al. (2019)) and may be due to a minor variation in the performances of one of the experimental measurement systems while changing the set-up.

– Last point in Section 4.2 - Authors state that "the thrust gain for a 30° yaw angle compared to the 0° case is around 13%, irrespective of the flow conditions" is a bit misleading. It is not immediately clear that the thrust gain and the yaw angle are for different turbines. This is also true for the right subfigure of Figure 5. The axis labels are a bit confusing. at first glance, it looks like a ratio of the thrust coefficient of a wind turbine under yaw to that of the same turbine without yaw.

Update labels to make more clear.

The authors apologise but they don't see the ambiguity of the statement. Indeed, taking into account of the start of the section 4.2 "Figure 5b shows the downstream WT thrust coefficient versus the static yaw angle applied to the upstream WT...", together with the definition of $C_T$ (title of section 3.3), the authors thought that the reader had all the elements to understand the figure. For the sake of completeness the sentence has been modified as it follows:

the thrust gain of the downstream WT for a 30° yaw angle variation of the upstream WT compared to the 0° case is around 13%, irrespective of the flow conditions

– Figure 5 - These figures would benefit from error bars. Maybe place cases side-by-side in groups for each value of $\gamma$?

The chosen representation of the figures is the clearest result of different tests. Regarding the error bars, due to the different values according to the set-ups, it has been chosen to detail the errors in the text ( sections 3.1 and 4.1)

– Equation 4 - For clarity, authors should specify that for positive yaw maneuvers, $\gamma_{start} = 0$, and for negative yaw manoeuvre, $\gamma_{end} = 0$

The sentence has been modified as it follows:

with $\gamma_{start}$ and $\gamma_{end}$ the $\gamma$ values before and after the transient. It has to be noted that for positive yaw manoeuvre, $\gamma_{start}$ = 0, and for negative yaw manoeuvre, $\gamma_{end} = 0$.

– It would be informative to see higher order statistics of the key phenomena of interest. PDFs of wake deviation angle and thrust coefficient would help readers and researchers understand that these are stochastic quantities.

Actually, it is not possible to retrieve the PDFs of the wake deviation angle and thrust coefficient. Indeed, these parameter are obtained after a phase-averaging of the velocity profile or the balance measurement respectively. As example, although for PIV measurement, hundreds of image pairs were acquired, the wake center has been calculated by processing the phase-averaged velocity field (similar approach for the thrust measurements).

– If the fit functions in Eqs. (7) and (8) are applied to the estimated yaw center and change in thrust of the downstream turbine, shouldn't $\tau_{lag}$ represent the convective time between the upstream turbine and either the measurement location or the downstream turbine? Also, the transient dimensionless duration is missing 10% of the fit, and thus will systematically underestimate the actual duration. This will lead to misleading values of $\Delta\tau*_{ratio}$.

In order to answer these questions, a new section 5.4 has been added to the paper. It explains that $\tau_{lag}$ can indeed be interpreted as the convection time and $c$ as the transient duration. A discussion on the consequence of the use of thresholds to determine the start and end is also added.

– The measured overshoot of thrust coefficient is not modeled or discussed at all (see Figure 7). This seems like an important physical element of this phenomenon. Consider modifying the models in Eqs. (7) and (8) if the overshoot appears consistently. Are the curves of $C_T$ in Figure 7 averages over many maneuvers?

Yes they are. The $C_T$ curves in Fig. 7 are averages over many manoeuvre and moreover, the averages are subjected to a filtering approach. Indeed, as explained in section 2.3, the phase averaging itself was not sufficient to properly filter out the load fluctuations due to the balance resonance. It is true that for the example case in figure 7, an overshoot of the thrust coefficient is visible and not discussed. It is indeed difficult to state whether this overshoot belongs to the actual dynamic response of the model to the load variation, or whether it is a consequence of the filtering strategy tuning approach. The fact that the balance resonance frequency was relatively close to the frequency range of interest prevents the authors to draw a final conclusion. Authors decided, driven by the duration metrics assessment, to tune the filtering approach in order to limit its effect on the ramp slope, despite the residual overshoot at the edges (cut out by the threshold approach). The following sentence has been added in section 2.3.

The filtering tuning was driven by the duration metrics assessment (§ 5.1), in order to limit its effect on the ramp slope, despite the residual overshoot at the edges.

– Tables 3 and 4- Do these values represent least-squares fits to average values, or are the instantaneous data fit and fit values averaged afterward? The tables would benefit from higher order statistics or measurements of uncertainty.

The values in table 3 and 4 represent the least-square fits applied to average values. For this reason, it is not foreseen to show higher order statistics.

– page 14, line 288 - I think $\tau_m$ should be $\Delta\tau_m$

The reviewer is right, the mistake has been corrected.

– Figure 8 needs legends and should use different symbols as figures above, as the points represent different groupings of the test cases.

Actually the aim of figure 8 is to give an ensemble evaluation of the duration without focusing just on disc porosity and kind of manoeuvre. Authors decided on purpose not to use different symbols or additional legend, which would have "polluted" the figure. Anyway the detailed parameters for each case can be found in tables 3 and 4.

– Authors state that ""Theoretically, the fitting coefficient $\tau*_{lag}$ lag (Tables 3 and 4) can be interpreted as a time delay before the transient starts. Unfortunately, there is no clear relationship between this parameter and the $\tau_{start}*$ start. This illustrates the difficulty of capturing the actual transient start for the present study." It seems like $\tau_{start}$ should be $\tau_{lag}$ plus the time required for the function to reach 0.5 based on the parameter c.

Regarding the fitting coefficients interpretation, a new section (section 5.4) has been added to the paper, dealing explicitely with this topic. Therefore the statement reported by the reviewer has been modified as it follows:

[revised manuscript text omitted]

**Minor points:**

– page 1 - paper

– page 1 - which is increasingly studied

– page 1 - started to be envisaged.

– page 1 - Remove "ity"

– page 1 - Remove the work "in" from XXXX

– page 6 - Table 2 - units should be written in normal text rather than math font.

– page 6 - Remove ical

– page 6 - kHz

– page 6 - "performed in order to perform" is redundant. Please rephrase. "designed in order to perform"

– page 6 - Text subscripts should be changed (e.g. $T_{filtered}$ instead of $T_{filtered}$)

– page 7 - Assumptions are made by the authors in fact. Replace "can be" with "is".

– page 8 - treated - considered?

– page 8 - it is not clear what 'usual error propagation methods' are. Please be specific.
The sentence has been modified as it follows:
As concerns measurement uncertainties, taking into account the uncertainties on the wind speed, the PIV vector resolution (see Table 2) and the methods used to estimate the wake center, it was possible to estimate the maximal measurement

error for both set-ups by applying usual resolution-based error estimation methods. The estimated measurement uncertainty is $\theta=\pm0.07°$ & $Y_c = \pm4.2\times10^{-4}m$ for HIT1 conditions and $\theta=\pm0.04°$ & $Y_c = \pm6.5\times10^{-4}m$ for HIT2 and ABL conditions.

The minor points listed by the reviewer have been taken into account.

---

## Author Comment (AC3) · 4 Feb 2021

**Reviewer comments**

Stefano Macrì, Sandrine Aubrun, Annie Leroy, Nicolas Girard

Co-authors thank the reviewers for their fruitful comments and will give point by point answers below. In this response to the reviewers the response of the authors not included in the revised manuscript will be written in blue while the part included in the revised manuscript will be written in red.

**1  Response to reviewer 2**

The paper provides an experimental study of certain aspects of yaw dynamics. The experimental approach seems fine and the writing is generally clear. My main concerns lie in the contribution of the work in terms of knowledge about what the authors call wake deviation dynamics and the loading on downstream turbines, which does not seem to be reported (discussed). The conclusions and discussion of the results lack a connection to the physics, which would potentially yield the insights that the paper aims to provide.

I was confused by the term wake deviation (dynamics) as it is not precise and was not really clarified in the paper.

The term *wake deviation* is used to describe the fact that, when a wind turbine is misaligned, its overall wake deviates from its normal position. Its normal position is given for a yaw angle equal to zero (rotor normal to the wind direction and so, wake centerline aligned with the wind direction). In this study, a wake deviation angle is considered to measure the wake deflection. *Finally, wake deviation dynamics* refers to the fact that one wants to study the dynamics of this phenomenon, through the analysis of the transient aspects of this wake deviation process during yaw manoeuver. More particularly, in order to clarify the term wake deviation, the following sentences have been added or rephrased to the revised paper :

in the abstract

This article investigates the far wake response of a yawing upstream wind turbine and its impact on the global load variation of a downstream wind turbine. In order to represent misalignment and realignment scenarios, the upstream wind turbine was subjected to positive and negative yaw manoeuvres.

While wake flow and wind turbine load modifications during yaw manoeuvres are usually described by quasi-static approaches, the present study aims at quantifying the main transient characteristics of these phenomena.

and in the introduction

deviate its wake from its nominal position

response during the yaw manoeuvre of a wind turbine

PIV fields aim at measuring the profile for the far-wake velocity distribution and permit to deduce the wake deflection. The latter can be described by a wake skew angle that depends on the yaw angle and the downwind distance in particular (Bastankhah and Porté-Agel (2016)). In this study, a wake deviation angle, computed from the estimation of the wake center displacement will be considered. In addition, a wake deviation duration will be introduced to analyse the transient aspects of this wake deviation process during yaw manoeuvre.

It was not clear that the balance would detect asymmetric loading on the disk, which might be important in this context. In general little discussion of the downstream turbine was affected is provided.

The aerodynamic balance is a 6DoF balance and so, can measure the asymmetric loading (see Muller et al. (2015) in which this asymmetric loading measurement was used to characterise the wake meandering). The objective in the present study was to use the loading of the second wind turbine as an indirect indicator of the upstream WT wake deviation and of the induced downstream WT thrust increase. The analysis was focused on the thrust measurement because that is the only loading that can be related to a notion of WT performance. This second WT model is considered here as a sensor of the global WT performance variations. Additionally, the streamwise force (thrust here) has a much higher value than the crosswise forces; measurement accuracy is therefore higher in the streamwise direction. In order to clarify this statement, the following sentences have been added to the revised paper in section 2.3.

It should be noted that the aerodynamic balance can detect the asymmetric loading on the downstream WT. Nevertheless the analysis will be focused on the thrust measurement because that is the only loading that can be related to a notion of WT performance.

There are some additional yaw models not discussed in the literature review that should be included, particularly in terms of computing the behavior of the wake centerline and in capturing the yaw dynamics (which the authors claim has never been done). Given those works it is unclear that the wake deviation angle is a meaningful metric since the centerline is not a straight line due to the curled wake etc., discussed even in some of the papers cited.

The reference suggested by the reviewer has been added in the introduction and the following sentences have been rephrased to take the reviewer's remark into account.

Some studies on the effects of yaw misalignment on wind turbine wakes, mainly based on quasi-static approaches, have already been carried out describing the effect of WT yaw on the wake position, in wind tunnel conditions (e.g. Bastankhah and Porté-Agel (2016); Grant et al. (1997); Howland et al. (2016); Schottler et al. (2018)) and at full scale (e.g Howland et al. (2020)). However, analyzing yaw manoeuvre dynamics, by studying the transient process between the non-yawed and yawed conditions, affords new insights into wake interactions.

As concerns the second part of the reviewer comment, it is right. Indeed, one underlying conclusion of the present work is that the wake deviation is particularly difficult to quantify because (i) the wake deviation angle is one angle of magnitude smaller than the yaw angle and because (ii) the definition of the wake center is not clear for an asymmetric wake distribution as encountered for skewed wakes. Therefore, the wake center position is not the best candidate to characterise the consequences of the wake modification during yaw manoeuvre. The more global indicators, as the available wind power for a downstream wind turbine or the thrust applied to downstream wind turbine, give better and less ambiguous results.

I was somewhat confused by the choice of metrics in general, particularly in section 5.1. There was little discussion given as to why these are the correct metrics to consider in practice. The chosen ones for example do not provide information about what the downstream turbine might see in terms of velocity, wake or loading (although these things were alluded to in the abstract). These issues are of great importance and of far greater interest than the effect of cyclic yawing behaviour, which is not clear is common in practice.

The chosen metrics actually provide information on the yaw angle of the upstream WT, the wake deviation angle at the location of a downstream WT and the loading applied to the downstream WT. The present paper is focused on the determination of the transients (dynamical part of the processes), which are generally described, as a first approach, by their duration and the time delay between the action (yaw manoeuvre) and reaction (wake skewing for instance). Consequently, it has been decided to use normalised metrics to restrain the attention to these transient parameters. The reviewer refers to the cyclic yawing. This principle is indeed not used in practice. This is more a "trick" that is used in the present wind tunnel study to be able to reproduce automatically 500 times the same yaw manoeuvre and then to obtain converged statistics by phase-averaging velocity and load measurements. Each cycle is composed of a yaw increase from 0° to 30°, a still period, a yaw decrease from 30° to 0°, a still period. During the result analysis, yaw increase transients and yaw decrease transients are studied separately.

The choice of configuration was also strange since the streamwise spacing was short for all cases (in practical wind farms spacings are often at least 5-7D) and there was little justification for these choices.

The reviewer is right when saying that the spacing is quite small compared to real wind farms. But, since the present work falls within a set of previous and ongoing studies for which the layout of the Ablaincourt-Pressoir wind farm operated by Engie Green was already used as reference, the same spacing as for this wind farm had been chosen. The wake interactions of two specific wind turbines were particularly investigated (Garcia et al. (2019); Macrì et al. (2020)) and some yaw control strategies have been tested on these both WT. In order to justify these choices, the following sentence has been added in the revised paper in section 2.1.

The fixed downstream distance was chosen according to the data set from a working wind farm studied in Garcia et al. (2019) & Macrì et al. (2020).

The list of results in bullet form was rather superficial and were merely observations of the data rather than an analysis. The presentation of the results as a list has been modified (see green part).

Why are equations (7) and (8) the appropriate choice; they just empirical fits to a chosen shape but the significance of this shape or choice is not mentioned.

The reviewer raises a relevant question that could be investigated in another future study. A parametric study on the type of fit functions was performed before ending-up with the chosen ones. Exponential laws were used because first, they fit well to the measured data and also because they provide tunable parameters that can be physically interpreted : $c$ can be straightfor- wardly associated to the transient speed and $\tau_{lag}$ to the time delay. A section (5.4) dealing with the fitting coefficient results interpretation has been added (see response to reviewer 1).

A few minor comments

Using S for area is unnecessarily confusing.
$S_D$ is replaced with $A_D$ in the revised paper.

Figure 1 should be larger (there is a lot of white space so it should be easy to make it easier to read and see all of the detail within the sketch).
The figure 1 has been reworked following this comment.

**References**

Bastankhah, M. and Porté-Agel, F.: Experimental and theoretical study of wind turbine wakes in yawed conditions, Journal of Fluid Mechanics, 806, 506–541, 2016.

Garcia, E. T., Aubrun, S., Coupiac, O., Girard, N., and Boquet, M.: Statistical characteristics of interacting wind turbine wakes from a 7-month LiDAR measurement campaign, Renewable energy, 130, 1–11, 2019.

Grant, I., Parkin, P., and Wang, X.: Optical vortex tracking studies of a horizontal axis wind turbine in yaw using laser-sheet, flow visualisation, Experiments in fluids, 23, 513–519, 1997.

Howland, M. F., Bossuyt, J., Martínez-Tossas, L. A., Meyers, J., and Meneveau, C.: Wake structure in actuator disk models of wind turbines in yaw under uniform inflow conditions, Journal of Renewable and Sustainable Energy, 8, 043 301, 2016.

Howland, M. F., González, C. M., Martínez, J. J. P., Quesada, J. B., Larranaga, F. P., Yadav, N. K., Chawla, J. S., and Dabiri, J. O.: Influence of atmospheric conditions on the power production of utility-scale wind turbines in yaw misalignment, Journal of Renewable and Sustainable Energy, 12, 063 307, 2020.

Macrì, S., Duc, T., Leroy, A., Girard, N., and Aubrun, S.: Experimental analysis of time delays in wind turbine wake interactions, in: Journal of Physics: Conference Series, vol. 1618, p. 062058, IOP Publishing, 2020.

Muller, Y.-A., Aubrun, S., and Masson, C.: Determination of real-time predictors of the wind turbine wake meandering, Experiments in Fluids, 56, 53, 2015.

Schottler, J., Bartl, J. M. S., Mühle, F. V., Sætran, L. R., Peinke, J., and Hölling, M.: Wind tunnel experiments on wind turbine wakes in yaw: redefining the wake width, 2018.